

# Assessing the characteristics and drivers of compound flooding events around the UK coast

Alistair Hendry[1], Ivan D. Haigh[1], Robert J. Nicholls[2], Hugo Winter[3], Robert Neal[4], Thomas
Wahl[5], Amélie Joly-Laugel[3] and Stephen E. Darby[6]

[1] Ocean and Earth Science, University of Southampton, National Oceanography Centre Southampton, University of Southampton, Waterfront Campus, European Way, Southampton, SO14 3ZH, United Kingdom.

[2] School of Engineering, University of Southampton, Highfield, Southampton, SO17 1BJ, United Kingdom.

[3] Natural Hazards and Environmental Group, EDF Energy R&D UK Centre, London, UK

[4] Department of Weather Science, Met Office, Exeter, UK

[5] Civil, Environmental, and Construction Engineering, National Center for Integrated Coastal Research, and Sustainable Coastal Systems Cluster, University of Central Florida, 12800 Pegasus Drive, Suite 211, Orlando, FL 32816-2450, USA.

[6] Geography and Environmental Sciences, University of Southampton, Highfield, Southampton, SO17 1BJ, United Kingdom.

Corresponding author: Alistair Hendry (A.Hendry@soton.ac.uk)

Submit to Hydrology and Earth System Sciences (**HESS**)

December 2018





# Abstract

In low-lying coastal regions, flooding arises from oceanographic (storm surges plus tides and/or waves), fluvial (increased river discharge) and/or pluvial (direct surface runoff) sources. The adverse consequences of a flood can be disproportionately large when these different

sources occur concurrently, or in close succession, a phenomenon that is known as 'compound flooding'. In this paper, we assess the potential for compound flooding arising from the joint occurrence of high storm surge and high river discharge around the coast of UK. We hypothesize that there will be spatial variation in compound flood frequency, with some coastal regions experiencing a greater dependency between the two flooding sources than others. We

map the dependence between high skew surges and high river discharge, considering 326 river stations linked to 33 tide gauge sites. We find that the joint occurrence of high skew surges and high river discharge occurs more frequently during the study period (15-50 years) at sites on the south-west and west coasts of the UK (between 3 and 6 joint events per decade), compared to sites along the east coast (between 0 and 1 joint events per decade). Second, we investigate

the meteorological conditions that drive compound and non-compound events across the UK. We show, for the first time, that spatial variability in the dependence and number of joint occurrences of high skew surges and high river discharge is driven by meteorological differences in storm characteristics. On the west coast of the UK, the storms that generate high skew surges and high river discharge are typically similar in characteristics and track across

the UK on comparable pathways. In contrast, on the east coast, the storms that typically generate high skew surges are mostly distinct from the types of storms that tend to generate high river discharge. Third, we briefly examine how the phase and strength of dependence between high skew surge and high river discharge is influenced by the characteristics (i.e. flashiness, size, elevation gradient) of the corresponding river catchments. We find that high

skew surges tend to occur more frequently with high river discharge at catchments with a lower base flow index, smaller catchment area and steeper elevation gradient. In catchments with a high base flow index, large catchment area and shallow elevation gradient the peak river flow tends to occur several days after the high skew surge. The previous lack of consideration of compound flooding means that flood risk has likely been underestimated around UK coasts,

particularly along the southwest and west coasts. It is crucial that this is addressed in future assessments of flood risk and flood management approaches.

**Key words:** Compound flooding, coastal flooding, fluvial flooding, storm surge, river discharge, UK.





# 1. Introduction

Flooding is the most dangerous and costly of natural disasters (Pall et al., 2011). From 1980 to 2013, floods accounted for more than $1 trillion in losses and resulted in at least 220,000 fatalities globally (Munich Re, 2017). More than 50% of these deaths and a large proportion of the economic losses occurred in densely populated low-lying coastal regions. Globally, coastal areas are home to more than 600 million people and constitute strategic economic centres (McGranahan et al., 2007). Recent flood events, for example, Cyclone Nargis in Myanmar (Fritz *et al.*, 2009), Hurricane Katrina in the US (Jonkman *et al.*, 2009), flooding in the UK over the winter of 2013/14 (Haigh et al., 2016), and Hurricane Harvey in the US (Emanuel, 2017) have demonstrated the ever-present threat of serious flood impacts in coastal regions despite improvements in levels of flood protection and advancements in flood forecasting and warnings. Furthermore, coastal flooding is a growing threat due to sea-level rise and changes in storminess (Church et al., 2013), ongoing vertical land movement (Brown and Nicholls, 2015), and rapid population growth and accompanying development in flood-exposed areas (Hallegatte *et al.*, 2013).

Flooding in coastal regions arises from four main source mechanisms: (1) storm surge combined with high astronomical tide (storm-tides); (2) local or remotely (swell) generated waves; (3) river discharge (fluvial); or (4) direct surface runoff (pluvial). The first two sources are oceanographic in origin, while the latter two mainly arise from heavy precipitation, but can also arise from snow melt. Most existing flood risk assessments consider these four main drivers of flooding separately. However, in coastal regions floods are often caused by more than just one factor, because they may be naturally correlated (i.e., with storms). Furthermore, the adverse consequences of a flood can be greatly exacerbated when the oceanographic (storm-tides and waves), fluvial, and/or pluvial drivers occur concurrently or in close succession (i.e., a few hours to days apart). Depending on local characteristics (which influence lag times between variables), this can result in disproportionately extreme events, referred to as 'compound flood events'. Compound events are defined by the Intergovernmental Panel on Climate Change (Seneviratne *et al.*, 2012) as: *(1) two or more extreme events occurring simultaneously or successively; (2) combinations of extreme events with underlying conditions that amplify the impact; (3) combinations of events that are not themselves extremes but lead to an extreme event when combined.* Zscheischler et al. (2018) define compound events as *the*



*combination of multiple drivers and/or hazards that contributes to societal or environmental risk*. With the potential to create considerable destruction, the World Climate Research Program (WCRP) Grand Challenge on Extremes has recently identified compound events as an international research priority (Zscheischler *et al.*, 2018). In this paper we assess the

5    potential for compound flooding arising from the joint occurrence of high storm surges and high river discharges around the coast of UK.

A recent example of compound flooding occurred during Hurricane Harvey in 2017. Record breaking rainfall, river discharge and runoff, combined with a moderate but long-lasting storm

10   surge, resulted in disastrous flooding in Houston (Emanuel, 2017). It was the second costliest natural disaster in US history (NOAA, 2018). Hurricane Irma in 2017 was also a prime example of compound flooding, where significant flooding occurred along the St. Johns River in Jacksonville, as a result of a combined storm surge and extensive rainfall runoff (Cangialosi, Latto and Berg, 2017). Compound flooding can also arise from extra-tropical storms. For

15   example, a storm surge on the Adriatic coast of Italy obstructed large amounts of freshwater runoff (generated by the same storm) from draining, causing major compound flooding in Ravenna, Italy (Bevacqua *et al.*, 2017). It is now recognised that by not considering compound flooding, the risk to these locations and elsewhere had been, and continues to be, greatly underestimated (Wahl et al., 2018).

In recent years there has been an increase in the number of studies that have started to investigate compound flood sources and events. The majority of these studies have been undertaken on a small spatial scale for specific localised regions, e.g., Fuzhou City, China (Lian *et al.,* 2013); Tsengwen River basin, Taiwan (Chen and Liu, 2014); Hudson River, USA (Orton

25   *et al.*, 2015); Shoalhaven River, Australia (Kumbier *et al.*, 2018); the Rhine delta, Netherlands (Kew *et al.*, 2013 and Khanal *et al.*, 2018); Brest, France (Mazas and Hamm, 2017); Santander, Spain (Rueda *et al.*, 2016); Ravenna, Italy (Bevacqua *et al.*, 2017) and the River Trent, the Yare Basin, the River Ancholme, the River Taff and Lewes, East Sussex in the UK (Granger, 1959; Mantz and Wakeling, 1979; Thompson and Law, 1983; Samuels and Burt, 2002; and

30   White, 2007, respectively). These studies have typically examined the dependence between two source variables only, such as storm surge (or storm-tide) and river discharge, between storm surge and waves, or between storm surge and rainfall (as a proxy for runoff). Larger-scale assessments of compound flood events have been undertaken more recently for Australia (Zheng et al., 2013 and Wu *et al.*, 2018), the USA (Wahl et al., 2015), the UK (Svensson and



Jones, 2002, 2004) and Europe (Petroliagkis *et al.,* 2016; Paprotny *et al.,* 2018). Recently, Ward et al. (2018) assessed the dependence between coastal and river flooding on a quasi-global scale, using observational datasets.

5   This paper focuses on the UK, where coastal flooding is ranked as the second highest risk for causing civil emergency in the Government's National Risk Register (Cabinet Office, 2015). A series of studies in the 1990s and early 2000s, commissioned and funded by the Department for Environment Food and Agricultural Affairs (DEFRA), examined the dependence between coastal and river flooding around the UK coast (e.g., Hawkes *et al.*, 2002; Svensson and Jones,

10   2002, 2004; Hawkes and Svensson, 2003; Hawkes, 2005; Hawkes, Svensson and Surendran, 2005). These investigations found that large storm surges are more likely to coincide with high river discharge events at sites on the west coast than the east coast of the UK. Petroliagkis *et al.* (2016) analysed the dependence between storm surge, wave height and river flow at selected sites around Europe, including a few sites in the UK. More recently, Paprotny *et al.* (2018)

15   examined the dependency between storm surge, river discharge and rainfall across Europe including a greater number of sites in the UK than had previously been analysed. They both identified similar spatial patterns in the strength of dependency between storm surges and river discharge across the UK. In their global study, Ward et al. (2018) also identified a west/east difference in the strength of dependence between storm surge and river discharge for the UK.

20   However, none of these studies identified the reason(s) for this spatial variability.

In this paper, we build on these earlier studies and assess the potential for compound flooding arising from the joint occurrence of high storm surge and high river discharge around the coast of UK. We have three specific objectives that seek to advance the earlier studies. The first

25   objective is to map the spatial dependence between storm surges and high river discharge around the UK, comparing different methods for quantifying the dependence between these two variables. The research question is where do compound flood events occur around the coast? Our hypothesis is that there will be a spatial variation in compound flood frequency, with some coastal regions experiencing a greater dependency between the two flooding sources

30   than others.

Our second objective is to investigate the meteorological conditions that drive compound and non-compound events across the UK. Svensson and Jones (2002, 2004) briefly examined storm tracks associated with joint occurrence events, but here we undertake a much more extensive



meteorological analysis, and use the results from this to explain, for the first time, why large storm surges and high river discharge are more likely to coincide on the west UK coasts, compared to the east coast. The research question is which weather types favour the occurrence of compound events? Our hypothesis is that certain types of weather conditions will favour the

joint occurrence of storm surge and river discharge while other weather patterns will not.

Our third and final objective is to briefly examine how the strength and phase of dependence between storm surge and river discharge are influenced by the characteristics (i.e., flashiness, size, elevation gradient) of the corresponding river catchments. We hypothesize that the lower

the flashiness, the smaller the catchment area and the greater the average elevation gradient, the more likely that storm surges will occur around the same time as high river discharge.

The structure of this paper is as follows: Sections 2 and 3 describe the data and methods, respectively. The results are described in Section 4 and key findings are discussed in Section

5. Conclusions are given in Section 6.

## 2. Data

We used four main data types in this study, namely: (1) sea level time-series; (2) river discharge records; (3) meteorological datasets; and (4) river catchment characteristics. Those are

described in the following four sub-sections. In Section 2.5 we then describe how we select the tide gauge and river discharge sites for the subsequent analysis outlined in Section 3.

### 2.1. Sea level data

Sea level time-series from the UK National Tide Gauge network were obtained from the British

Oceanographic Data Centre (BODC; https://www.bodc.ac.uk). Data is available for 42 tide gauge sites around the UK coast, but we focus on 33 sites (see Section 2.5). Sea level records are available as hourly measurements before 1993 and quarter hourly after 1993. The longest sea level record (Newlyn) starts in 1915, whilst the shortest (Portrush) begins in 1995. We consider data up to the end of 2016. The data has been previously quality controlled by the

BODC, with questionable values flagged as improbable, null or interpolated. Any values that were flagged as improbable or null have been removed from the analysis.



### 2.2. River discharge data

River discharge data was obtained from the UK's National River Flow Archive (NRFA; ttp://nrfa.ceh.ac.uk/). Data is available for more than 1,500 river gauge sites, but we focus on 326 sites (see Section 2.5). The measurements are available as daily mean rates. The longest

river discharge record (Kingston, on the Thames) starts in 1883, whilst the shortest (Deerhurst on the Severn), begins in 1995. Again, we consider data up to the end of 2016. The data has been previously quality controlled by the Centre for Ecology & Hydrology (CEH), and we excluded data that was flagged as suspect.

### 2.3. Meteorological data

We use gridded mean sea level pressure (SLP), near-surface U and V wind and precipitable water content (PWC, entire atmosphere considered as a single layer) fields to investigate the meteorological conditions that drive compound and non-compound events. The first two variables are chosen because they are the primary variables leading to storm surges, whereas

the latter is used as a proxy for rainfall. We use data from the 20[th] Century global meteorological Reanalysis, Version 2c (Compo *et al.*, 2011), obtained from the National Oceanic and Atmospheric Administration website (NOAA; https://www.esrl.noaa.gov/psd/data/20thC_Rean/). The fields have a spatial and temporal resolution of 2 degrees and 6 hours, respectively and are available from 1851. We focus on the

data within the area 34 ˚N to 70˚N and 60˚W to 20˚E, which encompasses the region where storms affecting the UK are generated and influence the region.

### 2.4. Catchment characteristics

We obtained or calculated river catchment characteristics from information on the NRFA

website, for each of the river discharge sites we analysed. We consider three catchment characteristics as follows: (i) base flow index (BFI); (ii) catchment area; (iii) and catchment elevation variation. The BFI is a measure of the proportion of the river runoff that derives from stored sources (Gustard, Bullock and Dixon, 1992) and gives an indication of the flashiness (how quickly a river responds to precipitation) of a catchment. The more permeable the rock

and soils in a catchment, the higher the base flow. Rivers draining impervious clay catchments (with minimal lake or reservoir storage) typically have baseflow indices in the range 0.15 to 0.35, whilst chalk streams have a BFI greater than 0.9 as a consequence of the high groundwater component in the river flow. The catchment area is the size of the drainage basin of a particular





river. Both the BFI and catchment area are provided directly on the NRFA website, for each catchment. The catchment elevation variation is a measure of the steepness of a catchment. The NRFA provide statistics on the elevation of the minimum and maximum elevations in a catchment, along with the elevations at the 10th, 50th and 90th percentiles of the river catchment.

5  We calculated an elevation variation index by taking the difference between the 90th and 10th elevation percentiles and normalizing these about the mean of all sites; values close to 1 indicate a catchment with a steep elevation gradient and values close to 0 indicate a catchment with a gentle gradient.

## 2.5 Site selection

From the available datasets, described above in Sections 2.1 and 2.2, we match combinations of tide gauge and river discharge sites that satisfied the following criteria: (1) there are at least 15 years of overlapping records; and (2) daily mean river discharge is at least 5 $m^3$/s at the river site. Previous studies often matched river gauge sites to the nearest tide gauge sties. However,

because of the complex topography of the coastline, this doesn't always associate a river gauge site to the hydrologically-relevant tide gauge (and coast) for that river system. Therefore, we visually matched each river site to the tide gauge site nearest to the appropriate river. Nine tide gauge sites ((1) Dover; (2) Newhaven; (3) Port Erin, Isle of Man; (4) St Hellier, Jersey; (5) St Marys, Isles of Scilly; (6) Stornoway, Lewis and Harris; (7) Lerwick, Shetland Islands; (8)

Lowestoft; and (9) Harwich), were excluded from the analysis as there was no appropriate nearby river systems with discharge measurements, or the corresponding overlapping record length was less than 15 years for that specific combination of sites.

Following this selection, there are 326 combinations of discharge stations and tide gauges, the

locations of which are shown in Figure 1, linked to 33 tide gauge sites. In Figure 1a, and subsequent figures of this nature hereafter, river sites discharging onto the west, east and south coasts of the UK are plotted as triangles, circles and squares, respectively. There is good spatial coverage across most of the country, except in the southeast. The river sites discharging along the southeast, tend to have discharges below 5 $m^3$/s, or the overlapping data lengths are less

than 15 years. Some tide gauge sites (e.g., Newlyn and Wick) are not in the near vicinity of where corresponding rivers drain into the sea. However, as storm surges have large spatial extents, they are close enough to be considered representative of the broader scale storm surge characteristics in that area.



The number of years for which overlapping data is available for both sites is also shown in Figure 1a. The tide gauge data was typically the shorter of the two sets. The mean overlapping length across all sites was 24 years, with a maximum of 50 years. Tide gauges had an average

of 10 river gauges linked to them (Figure 1b), with a minimum of 1 (Newlyn, Fishguard and Holyhead) and a maximum of 37 (Immingham). Details of the location of the combination of sites and their overlapping data lengths are given in Table 1.

## 3. Methods

The analysis was undertaken in three main stages, each addressing one of the three study objectives outlined above. These stages are described in turn in the sections below.

### 3.1. Joint occurrence and dependence

Our first objective is to map the dependence between storm surge and river discharge,

comparing different methods for quantifying the dependence between these two variables. For sea level we considered two parameters: (1) total still sea-level; and (2) storm surge (i.e., the meteorological component of sea level). To represent the latter, we used the skew surge parameter, which is the difference between the maximum observed high water and the maximum predicted (astronomical) high water, each tidal cycle, regardless of their timing. To

extract time-series of skew surges from the sea level records at each tide gauge site, we followed the approach of Haigh *et al.* (2016). To do this, we first undertook a harmonic analysis, for each calendar year, using the T-Tide harmonic analysis package (Pawlowicz et al., 2002) with the standard 67 tidal constituents. Each observed and predicted high water was identified, and the difference between the two was computed to give time-series of skew surges.

Daily maxima total still sea level and skew surge time-series were then extracted, at each tide gauge site. The exact time of the daily maxima was retained for the meteorological analyses, described later in Section 3.2. The river discharge records were obtained in the format of daily mean values, and so no pre-processing was necessary on these records.

Extreme levels were extracted for each of the three (i.e., total sea level, skew surge and river discharge) daily time-series, at each site, using a peaks over threshold (POT) approach. We used a declustering algorithm, with a storm length of 48 hours (which is appropriate as Haigh *et al.* (2016) found storms in the UK typically affect sea level for 3.5 days) to guarantee





independent events. We varied the threshold at each site to ensure that each of the three time-series had on average 2.3 to 2.5 extreme levels per year. This threshold range ensured that: (1) we had enough data points to estimate dependence between the variables reliably; and (2) the threshold was high enough for the exceedances to be considered 'extreme' (Svensson and

5 Jones, 2005). The average thresholds across all sites were the 99th, 99.1th, and 99.2th percentiles for total sea level, skew surge and river discharge, respectively.

We then used three different approaches to assess the dependence between total sea level or skew surge, with river discharge. The first approach we term hereafter the 'dependence

method'. Here, we measure dependence between the daily maximum total sea-level/skew surge and discharge time-series using Kendall's Rank Correlation $\tau$ (Kendall, 1938), which, unlike the Pearson's Correlation Coefficient, captures non-linear relationships. Significance was assessed at $\alpha = 0.05$ (i.e., 95% confidence level), using corresponding p values estimated from exact permutation distributions. We also repeat the analysis using time-lags from −5 to +5 days.

For example, for daily maximum skew surge we select corresponding daily maximum discharge values with time lags of −5, −4, −3, −2, −1, 0, +1, +2, +3, +4, and +5 days. This is to allow for that fact that when a storm approaches the coast, for example, it might first generate a high storm surge, before travelling inland and generating high precipitation and therefore elevated river discharge sometime afterwards.

The second approach we term hereafter the 'dependence extreme method' and it follows that previously applied by, for example, Wahl *et al*. (2015) and Ward *et al*. (2018). Here we start with the extreme total sea level, or skew surge values above the threshold, and then select the highest discharge values that occurred on the day of that event. This is referred to throughout

the paper as discharge conditional on high total sea-level or skew surge. We then start with the extreme river discharge levels above the threshold, and select the highest total water levels or skew surges that occurred on the day of that event. This is referred to throughout the paper as total sea level or skew surge conditional on high discharge. We again measure dependence between the total sea-level/skew surge and discharge time-series using Kendall's Rank

Correlation $\tau$ (Kendall, 1938). We also repeat the analysis using time-lags from −5 to +5 days.

The third approach we term hereafter the 'joint occurrence method'. Here we simply count the number of times extreme total sea levels events, or skew surges events, above the chosen





threshold for that site, occurred on the same day as extreme river discharge. Each pair of sites have varying overlapping data lengths. Therefore, to standardise the results, the number of joint occurrences per decade were determined. Again, we repeat the analysis but lag the discharge using time-lags of -5 to +5 days.

To illustrate the approaches, time-series of daily maximum skew surges are plotted against records of daily maximum river-discharge at 0-day lag for Devonport (southwest coast) and Whitby (east coast) in Figure 2a and 2b, along with the two percentile thresholds. At Devonport there are 9 occasions (red dots in Figure 2a) when extreme skew surges occur on the same date

as extreme river discharges, whereas at Whitby (Figure 2b) there are no coincident events.

## 3.2. Meteorological analysis

Our second objective is to investigate the meteorological conditions that drive compound (i.e., joint occurrence of high skew surges and large river discharge) and non-compound events (i.e.,

high skew surge or high river discharge only) events across the UK. For each site, we extract fields of SLP, wind speed and PWC for the 6-hour period closest to the peak of each: (1) extreme total sea level or skew surge event (i.e., all the events in zone 1 on Figure 2); (2) each joint occurrence event (i.e., all the events in zone 2 on Figure 2); and (3) each extreme river flow event (i.e., all the events in zone 3 on Figure 2). For each site, and each of these three

types of events, we derive composite plots of SLP, wind speed and PWC, by taking an arithmetic mean and standard deviation of the data at each hindcast grid cell, through the time of the corresponding events. The composite plots thus represent the mean (with variance around the mean) conditions of the storms that generate compound and non-compound events. We also digitised the tracks of all responsible storms, for the three different event types, using

the storm tracking algorithm developed by Haigh *et al*. (2016). This captures the location of the storm centre for each 6-hourly time step of the metrological reanalysis, from cyclogenesis to storm dissipation or when the storm leaves the area of interest (defined above). We calculate the mean storm track for each event type, at each site. This allows us to compare and contrast the weather patterns related to the storms which caused the compound and non-compound

events.



### 3.3. Catchment correlations

Our final objective is to briefly examine how the strength and phase of dependence between total sea level or skew surge and river discharge is influenced by the characteristics of the corresponding river catchments. To do this we calculate correlation coefficients between the strength of dependence (or number of joint occurrences per decade) and the maximum phase lag, with our three selected catchment variables (BFI, catchment area and catchment elevation variation). Again, significance was assessed at $\alpha = 0.05$. We hypothesize that the lower the BFI, the smaller the catchment area and the greater the average elevation gradient, the more likely that high total sea levels or skew surges will occur around the same time as high river discharge. The higher the BFI, the larger the catchment area and the gentler the elevation gradient of the catchment, the more likely it is that high river discharge will occur several days after high total sea level or skew surge, for the sites closest to the coast.

## 4. Results

### 4.1 Dependence and joint occurrences

We used three methods to assess the dependence between high total sea level or high skew surge with high river discharge, across the 326 combinations of discharge stations and tide gauge sites. The results of the first method, the dependence method, are shown in Figures 3a and 4a for daily maximum total sea level and daily maximum skew surge, respectively, with daily maximum river discharge, for 0-day lag. As expected, there is generally greater dependence between skew surges and river discharge (Figure 4a), than between total sea level and river discharge (Figure 3a). This is because total sea levels are strongly influenced by the deterministic tidal component, around the majority of the coastline of the UK (Haigh *et al.*, 2016). Interestingly, the dependence is stronger for total sea levels for sites linked to tide gauges in the northern Irish Sea (e.g., Portrush and Bangor in North Ireland and Portpatrick in England and Millport in Scotland), and this is most likely because tidal range is small here, and not such a dominant factor on total sea levels, compared to other sites. A clear spatial variation in the dependence between high sea levels or skew surges with high river discharge is evident in Figures 3a and 4a. For many of the sites along the south-west and west coasts of the UK, $\tau$ typically ranges from 0.1 to 0.35 whilst along the east coast this drops to 0.0 to 0.15. The greatest dependence is found at river gauges linked to the Millport and Portpatrick tide gauges



in southwestern Scotland. The lowest dependence is located at river gauges near Cromer on the East Coast. Two river sites linked to the Bangor tide gauge in Northern Ireland show negative dependence.

We also calculated the dependence between daily maximum total sea level or skew surge and daily maximum river discharge using time lags from −5 to +5 days. The results for high skew surge and high river discharge are shown in Supplementary Figures S1a to S11a for all sites. Dependence is typically weak until -1-day lag. Interestingly the dependence is higher for +1 to +5-day lags, compared to -5 to -1-day lags.  This is illustrated in Figure 5 for the six river sites

closest to the tide gauges of Bournemouth, Devonport, Workington, Ullapool, Whitby and Cromer. The distributions are typically skewed to the right, and this is probably because river levels remain elevated for several days after a storm event. The lag day when there is the maximum dependence between daily maximum skew surge and daily maximum river discharge is shown in Figure 6a, for all sites. Interestingly, 42 inland sites (13% of the 326

sites) on the east coast have a maximum correlation at -1-day lag. The majority of the sites (188; 58%) have maximum correlation at 0-day lag. Sites on the south-west and west coast typically have maximum correlations between +1 and +5 days. 50 sites (15%) have maximum correlation at +1-day lag, 19 sites (6%) at +2-day lag, 21 sites (6.4%) at +3-day lag, 3 sites (1%) at +4-day lag and 3 sites (1%) at +5-day lag. The sites with maximum correlations at +4

and +5-day lag are mostly situation in the Severn River, which has a large catchment area (see Section 4.3).

We compared these results with those obtained using the other two methods we applied. The results from the second approach, the dependence extreme method, are shown in

Supplementary Figures S12a to S22a for river flow conditional on high skew surge, and in Figures S12b to S22b for skew surge conditional on high river discharge. The strength of dependence is typically weaker than for the daily maximum time-series. However, similar spatial patterns are evident with sites on the south-west and west coasts showing statistically significant dependence, whereas for sites on the east coast the dependence was not typically

statistically significant.

The results for the third method, the joint-occurrence method, are shown in Figures 3b and 4b for high total sea levels and high river discharge, and high skew surges and high river discharge, respectively, at 0-day lag. The spatial patterns are very similar to those of the daily dependence





results. For many of the sites along the south-west and west coasts of the UK, there are a higher number of joint occurrences between high skew surges and high river discharge (between 3 and 6 joint events per decade), than for sites along the east coast (between 0 and 1 joint events per decade). Sites with the largest numbers of joint occurrences (5 to 6 events per decade)

include river discharge sites linked to Millport, Workington, Mumbles, Devonport and Bournemouth tide gauges. There are several sites along the south-west and west coasts which show low (< 1 event per decade) or zero joint occurrences at 0-day lag. These include river discharge sites linked to tide gauges at Heysham and Portsmouth in England, Bangor in Ireland, Barmouth and Milford Haven in Wales, and Portpatrick, Ullapool and Kinlochbervie in

Scotland. Interestingly, there is large variation on a regional/local scale, particularly in areas which mostly have high numbers of joint occurrences. For example, at many sites around the Bristol Channel, the number of joint occurrences varies between 1 to 4 per decade, at river discharge sites less than 80 km apart.

At 61 out of the 326 sites (19%), there are no joint occurrence events between high total sea level and high river discharge (Figure 3a). At 169 (52%), 76 (23%), 17 (5%), 2 (1%) and 1 (0.3%) sites, there are between 0 and 1, 1 and 2, 2 and 3, 3 and 4, and >4 number of joint occurrences per decade, respectively, between high total sea level and river discharge. At 24 out of the 326 sites (7%), there are no joint occurrence events between high skew surges and

river discharge (Figure 4a). At 97 (30%), 97 (30%), 56 (17%), 31 (10%), 14 (4%) sites and 7 (2%), there are between 0 and 1, 1 and 2, 2 and 3, 3 and 4, 4 and 5, and >5 number of joint occurrences per decade, respectively, between high skew surges and high river discharge.

The lag day when there are the maximum number of joint occurrences between high skew

surge and high river discharge are shown in Figure 6b, for all study sites. The results are similar to those seen for the daily maximum dependence approach (Figure 6a). Inland sites on the east coast typically have a maximum number of joint occurrences at -1 to -3 days lag, whereas several sites on the west coast have a maximum number of joint occurrences at +1 to +5 days lag.

## 4.2 Meteorological analysis

We now investigate the meteorological conditions that drive compound (i.e., joint occurrence of high skew surges and large river discharge) and non-compound events (i.e., high skew surge





or high river discharge only) events across the UK. We focus here on skew surge, rather than total sea level, as the dependence between skew surges and river discharge is stronger. At each of the 326 river discharge sites, we have derived composite plots of SLP, wind speed and PWC, by taking an arithmetic mean and standard deviation of the data at each hindcast grid cell,

through the time of the events that have led to: (1) high skew surge events only; (2) joint occurrence events; and (3) high river discharge events only. To illustrate the results of this component we focus on two contrasting sites: (1) Devonport on the UK southwest coast where high storm surges and high river discharge have occurred at similar times in the past (Figure 2a); and (2) Whitby on the UK east coast, where high storm surges have never occurred (during

the period of record) at times of high river discharge (Figure 2b). Examples of 8 other sites are shown in Supplementary Figures S23 to S30.

Composite plots are shown in Figures 7 and 8 for Devonport and Whitby, respectively, for SLP (left column), wind speed (central column), and PWC (right column) for the events that had:

(1) high skew surge only (top row); (2) both high skew surge and high river discharge (middle row); and (3) high river discharge only (bottom row). The number of events recorded for each type is listed, and the average standard deviation (SD), across all grid cells, is also reported. The latter gives an indication of the spread of the spatial patterns across all the corresponding events (i.e., a low SD indicates that the storms across all events have very similar spatial

patterns).

At Devonport (Figure 7), the meteorological patterns in SLP are similar across the three event types. All three event types feature a low-pressure system to the northwest of Ireland (Figures 7a, 7d and 7g), with strong south-westerly winds affecting the southwest coast. As expected,

the wind speed is more intense along the south coast for the skew surge only (Figure 7b) and joint event types (Figure 7e), compared to the river discharge only events (Figure 7g). The differences in PWC patterns are more pronounced. There is low PWC over the southwest for the surge only events (Figure 67c), and higher PWC for the joint and river only event types (Figure 7f and 7j). The composite plot of PWC is characterised by a higher SD for surge only

events (e.g., there is more spread across the range of events) in comparison to the joint and river only event types.

In contrast, at Whitby, the meteorological patterns in SLP are very different across the two event types (note, no joint high skew surge and high river discharge were observed here)



(Figure 8), showing that the storms that lead to high skew surges are distinct from the storms that lead to high rainfall and therefore river discharge. For high skew surge only events, the storm centre is situated over Scandinavia (Figure 8a), producing strong north-westerly winds across the North Sea (Figure 8b). PWC is low for the entire east coast (Figure 8c). For high

river only events, a weaker low-pressure system is centred over central UK (Figure 8g). The wind speeds are therefore low on the East Coast (Figure 8h). However, the PWC is high over much of the UK.

The results for other sites are similar (Supplementary Figures S23 to S30). For sites on the west

coast of the UK, the storms typically have similar SLP characteristics between the three event types. Whereas, for sites on the east coast, the storms are more distinct.

We also digitised the tracks of the storms responsible for each of the three event types, at these two selected sites. These storm tracks are shown in Figure 9 for Devonport (upper panels) and

Whitby (lower panels). The mean storm tracks are overlaid, in each instance. At Devonport, the mean storm tracks are similar, moving in an easterly north-easterly direction and cross over the north, or just to the north of Scotland (Figures 9a, 9b and 9c). In contrast, at Whitby, the mean storm tracks for the high skew surge events and high river discharge events are very different. The mean storm track for the high skew surge events pass to the north of Scotland

(Figure 9d), while the high river only events cross central UK (Figure 9f).

### 4.3 Localised correlations

The analysis of weather types (described in Section 4.2 above) has helped to explain national

scale spatial variations in the occurrence of compound events (i.e., the west/east difference shown in Figure 4), but to understand variations locally we need to consider other variables. We therefore briefly assess here, how the strength and phase of dependence between skew surge and river discharge is influenced by the characteristics of the corresponding river catchments.

The three selected catchment characteristics (BFI, catchment area and catchment elevation variation) are plotted in Figures 10a, 10b and 10c, respectively. The river sites that drain onto the central south coasts have the greatest BFI (nearly 1, i.e. extremely porous chalk) (Figure



10a). Catchments are largest on the Severn River, the River Bann in Northern Ireland and the east coast of Scotland, whereas smaller catchments are found in Cornwall, West Scotland and around Weymouth (Figure 10b). The largest elevation variation is seen on the River Spey in Scotland and altitude variation is low across the east coast of UK between Immingham and

Dover (Figure 10c). Visually, there is no obvious strong spatial correlation between any of the three catchment characteristics (Figure 10) and either the rank correlation between daily maximum skew surge and daily maximum river discharge (Figure 4a) or the number of joint occurrences per decade between extreme skew surge and extreme river discharge (Figure 4b).

The rank correlation for daily maximum skew surge and daily maximum river discharge (at 0-day lag) is plotted against the three catchment characteristics for each site in Figures 11a, 11b and 11c. The day of maximum lag for the rank correlation is plotted against the three catchment characteristics for each site in Figures 11d, 11e and 11f. Corresponding correlation coefficients (CC) are listed in Table 2, first for all sites and then just the river sites closest to the 33 tide

gauge sites. There is a negative correlation (CC = -0.5, significant at 95%) between dependence and BFI. This is in line with our hypothesis that the lower the BFI of the site (e.g., the flashier the catchment), the more likely that high skew surges will occur around the same time as high river discharge. There is a statistically significant negative correlation (CC = -0.31) between dependence and catchment area. Again, this is in line with our hypothesis that high skew surges

are more likely to occur around the same time as high river discharge in small catchments. There is a weak but statistically significant positive correlation (CC = 0.16) between dependence and catchment altitude variation. Again, this is in line with our hypothesis that the steeper the catchment, the more likely that high skew surges will occur around the same time as high river discharge. The correlation is higher (CC = 0.34, significant at 95%) for just the

33 river sites closest to each tide gauge site. The correlations between the three catchments and the day of maximum lag are not as strong (Table 2; Figures 11d, e and f). There is a weak statistically significant correlation (CC = 0.21, significant at 95%) between day of maximum lag and BFI. Sites with larger BFI typically have larger positive lags. There is also a weak, statistically significant, correlation (CC = 0.11, significant at 95%) between day of maximum

lag and catchment area. Sites with large catchment area typically have larger positive lags.




## 5. Discussions

In this paper we have assessed the potential for compound flooding arising from the joint occurrence of extreme total water level or skew surge and river discharges around the coast of UK. Like earlier studies (i.e., Svensson and Jones, 2002, 2004; Petroliagkis *et al.*, 2016;

Paprotny *et al.*, 2018) we have identified that the joint occurrence of high skew surges and high river discharge occurs more frequently on the south-west and west coasts of the UK, compared to the east coast. However, here we have been able to show, for the first time, that this spatial variability is driven by meteorological differences in storm characteristics. On the west coast of the UK, the storms that generate high skew surges and high river discharge, are typically

similar in characteristics (i.e., there is a low-pressure system to the northwest of Ireland with strong south-westerly winds affecting the southwest coast) and track across the UK on comparable pathways. In contrast, on the east coast, the storms that typically generate high skew surges (i.e., when there is a low pressure over Scandinavia producing strong north-westerly winds across the North Sea) are distinct from the types of storms that tend to generate

high river discharge in this area (i.e., when there is a weaker low-pressure system over central UK).

We also identified, for the first time, relationships across the UK between the strength and phase of the dependence between high skew surge and high river discharge and the

characteristics of the corresponding river catchments. We find that high skew surges tend to occur more frequently with high discharge in catchments with a lower base flow index, smaller area and steeper elevation gradient. In catchments with a high base flow index, large area and shallow elevation gradient the peak river flow tends to occur several days after high skew surge. We also found that for inland river discharge sites on the east coast, the maximum number of

joint occurrences happens when river discharge occurs -1 days before peak skew surge. This is because the maximum storm surge in the North Sea occurs after the storm has crossed the North Sea into Scandinavia, whereas the high rainfall occurs a day earlier when the storm is centred over the UK.

The key concern for compound flooding is when estuaries or coastal regions experience both high storm surge and high river discharge around the same time (i.e., 0-day lag), which is likely to lead to disproportionately large adverse flood consequences. Of the 33 tide gauge sites considered, dependence between high skew surge and high river discharge is maximum at 0-



day lag at 19 sites (for the river discharge station closest to these sites; see Supplementary Figure S31). At most other sites, high river discharge occurs between +1 and +5 days after peak skew surge, and therefore compound flooding is not as much a concern. However, there are still important implications for flood management and emergency response if a large fluvial flood occurs several days after a major coastal flood, as this is likely to stretch emergency services.

The meteorological analysis we have undertaken, indicates subtle differences in the types of storms that tend to generate compound events, compared to non-compound events, particularly for sites on the west coast of the UK (see Figure 6). As compound events tend to exacerbate the adverse consequences of a flood, it is vital they are forecasted accurately, and appropriate warning is provided. Furthermore, the best response to a compound event might differ from a non-compound event. Therefore, to be able to accurately forecast that an event might be a compound event, as opposed to a non-compound event, is crucial. With these insights and improvement in forecast opportunities discussed below, these aspects of emergency response should be analysed in more detail.

The UK Flood Forecasting Centre (a collaboration between the Environment Agency and Met Office) have developed a medium- to long-range operational forecasting tool called Coastal Decider (Neal *et al.,* 2018). This is based on probabilistic weather-pattern forecasts and helps to identify periods with an increased likelihood of coastal flooding from high sea levels around the UK. Coastal Decider uses a set of 30 distinct weather patterns (referred to as the 'Met Office weather patterns') which were derived by Neal *et al.* (2016) using k-means clustering techniques. These weather patterns (shown in Supplementary Figure S32) represent the large-scale meteorological conditions experienced over the UK and surrounding European area. Neal *et al*. (2018) used a daily historical weather pattern catalogue to show that particular weather patterns tend to relate to high sea level events at different sites around the UK, with this analysis forming the basis for Coastal Decider. Other research which relates the Met Office weather patterns to meteorologically induced hazards includes Richardson *et al.* (2018), who related the weather patterns to precipitation observations for the application of drought forecasting.

Here, we use the same daily historical weather pattern catalogue as Neal *et al.* (2018) and Richardson *et al.* (2018) to calculate the modal weather pattern at each site for: (1) high skew surge events only; (2) joint occurrence events, and; (3) high river discharge events only. This





is done in order to briefly assess whether Coastal Decider could be expanded to give early warnings of events with the potential to generate compound flooding from both high sea level and high river discharge. Results are shown in Figure 12. Nearly all the events are dominated by the higher numbered weather patterns, which tend to be the more stormy types and which

are most likely to occur in the winter. Clear distinctions are found along coastal regions. Weather pattern 30 occurs for sites along the southwest and west UK coast, for each of the three types of events. This is one of the stormiest weather patterns with a large depression situated to the north of Scotland. This causes a strong westerly flow across the UK with frontal rainfall being particularly heavy in western parts of the UK. Weather pattern 20 is dominant

along the central west coast, particularly from the Bristol Channel northwards. This weather pattern is similar to weather pattern 30, but with the depression centre further north, therefore shifting the wind and rain impacts further north. Sites in Scotland typically feature weather patterns 20 (cyclonic westerly) and 21 (cyclonic south-westerly). Along the east coast high skew surge and river discharge events experience different weather patterns with pattern 14

(cyclonic northerly) generally being seen during high skew surge events compared to patterns 11 (low pressure centred over the UK), 24 (southerly tracking cyclone centred over the North Sea) and 30 (very cyclonic westerly) which are generally related to high river events. These results indicate that it may be possible to extend the forecasting capability of Coastal Decider to also include indications for the likelihood of compound events. Small scale weather features

will need to be included in the mean composites for each weather pattern (e.g., weather pattern 30, which is a very stormy cyclonic south westerly type, will have a mean composite that is formed from many subtle variations in the overall broad-scale stormy south westerly flow; this means that the small scale (and perhaps rarer) features will still be represented within a broader scale weather pattern).

So far, we have just considered high water levels which produce the potential for flooding. In periods of high run-off in the UK such as 1998, 2000 and 2007 floods happened repetitively near the tidal limit of rivers due to tidal locking at high tide such as the floods in Lewes in 2000 (White, 2007). However, these may not be compound events as defined here. To briefly assess

the extent of flooding during compound events, we compared the dates of joint occurrences at Devonport (which had a higher number of joint occurrences per decade) with reports of coastal flooding in the SurgeWatch database (Haigh *et al.*, 2015, 2017). SurgeWatch records the social, economic and environmental consequences of 330 coastal floods that have impacted the UK in the last 100 years. Of the 9 joint occurrence events when there was both high skew surge



and high river discharge observed at Devonport, 7 events had reports of coastal flooding. Events with significant flooding included: 24th-25th December 1999, which caused extensive flooding in Lymington, Dorset, as discussed below; and the 14th February 2014 storm which led to the destruction of the main railway line in Dawlish (Devon Maritime Forum, 2014) (note, this event also had large waves). No flooding was reported for the Great Storm of 15th-16th October 1987 (Burt and Mansfield, 1988). There was extensive wind damage during this event to the UK, but little coastal flooding because the event coincided with neap tides.

The 24th-25th December 1999 compound flooding event in Lymington is especially noteworthy and illustrates the need to consider compound events in the design of flood protection schemes. On the 16-17th December 1989 Lymington was flooded by high sea levels and waves, with considerable damage to 50 houses and the railway line (Ruocco *et al.*, 2011; Haigh *et al.*, 2015). This event was the driving force for a large upgrade of coastal flood defences for the town, including new sluice gates which allowed the Lymington River to drain at low tide, but sealed it from tidal flooding during high sea levels. However, no allowance or consideration of compound flooding appears to have been made at the time. Ten years later, on 24th December 1999, a storm generated a storm surge which did not directly cause flooding itself, because of the raised defences. However, the storm surge prevented the sluice gates from opening for prolonged periods, while large volumes of rainfall during the storm raised river flow. Combined with the lack of drainage, this caused flooding from the river on the upstream side of the sea defences (Ruocco *et al.*, 2011). Subsequently the Lymington flood defences were upgraded again. This event strongly highlights the importance of considering compound flooding when assessing and designing flood management.

## 6. Conclusions

This paper has assessed the potential for compound flooding arising from the joint occurrence of extreme total water level or skew surge and river discharges around the coast of UK. We found that the joint occurrence of high skew surges and high river discharge occurs more frequently during the study period (15-50 years) at sites on the south-west and west coasts of the UK (between 3 and 6 joint events per decade), compared to sites along the east coast (between 0 and 1 joint events per decade). We showed, for the first time, that the spatial variability in the dependence and number of joint occurrences of high skew surges and high river discharge is driven by meteorological differences in storm characteristics. On the west




coast of the UK, the storms that generate high skew surges and high river discharge are typically similar in characteristics and track across the UK on comparable pathways. In contrast, on the east coast, the storms that typically generate high skew surges are mostly distinct from the types of storms that tend to generate high river discharge. We found that high

skew surges tend to occur more frequently with high river discharge at catchments with a lower base flow index, smaller catchment area and steeper elevation gradient. In catchments with a high base flow index, large catchment area and shallow elevation gradient the peak river flow tends to occur several days after the high skew surge. The previous lack of consideration of compound flooding means that flood risk has likely been underestimated around UK coasts,

particularly along the southwest and west coasts. It is crucial that this is addressed in future assessments of flood risk and flood management approaches.

**Author contribution:** AH and IDH conceived, planned and carried out the analysis and

interpretations. RJN aided as an expert in coastal flooding. HW provided guidance on extreme value statistics. RN and AJL provided the knowledge on the meteorological component. TW offered his expertise in compound flooding. SED added his knowledge of fluvial processes. The manuscript was prepared by AH and IDH with contributions from all co-authors.

**Competing interests:** The authors declare that they have no conflict of interest.

**Data Availability:** This study relies entirely on publicly available data, described in Section 2. Dependence and joint occurrence results can be seen in Table S1 in the supplementary material.

**Acknowledgements:** We would like to thank EDF Energy R&D Centre UK and the Southampton Marine and Maritime Institute (SMMI) for funding this research. We would also like to thank the data providers: the British Oceanographic Data Centre; and the National River Flow Archive for supplying the sea level and river data, respectively. In addition, we'd like to

thank the Earth Systems Research Laboratory (ESRL, NOAA) for providing the 20[th] Century V2c meteorological reanalysis.



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





## Figures



**Figure 1:** (a) Location and overlapping data length (in years) of the 33 tide gauge sites (black dots) and 326 river discharge stations (triangles, circles and squares show the river stations that discharge onto the west, east and south coasts, respectively); and (b) pairing of the tide gauge and river discharge stations.




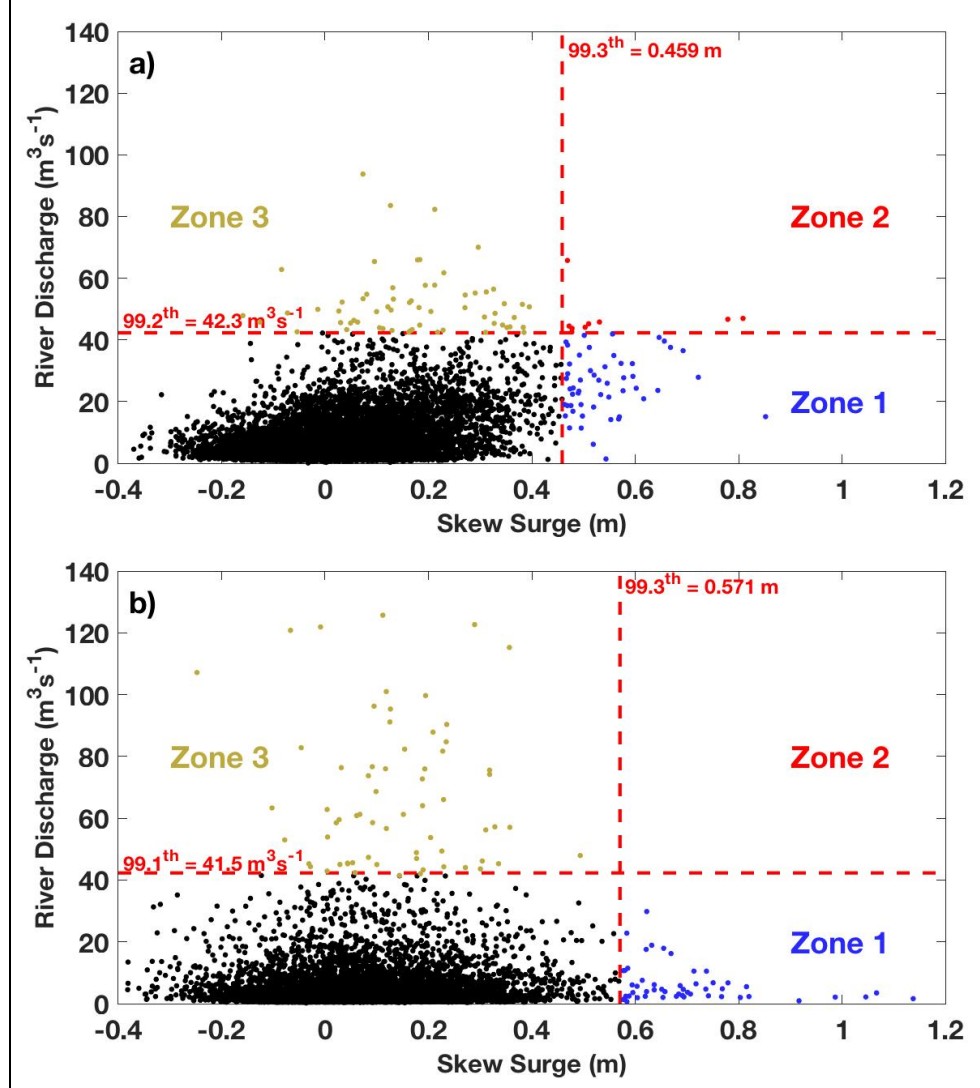

**Figure 2:** Daily maximum skew surge plotted against daily maximum river discharge for (a) Devonport; and (b) Whitby. The dotted red lines indicate the high percentiles chosen in the analysis for the two variables at these sites. Red dots (plotted in Zone 2) show the events with potential for compound flooding (i.e., joint occurrence of high storm surge and large river discharge) whereas blue (Zone 1) and green (Zone 3) define the non-compound events (i.e., high storm surge or high river discharge only, respectively).





**Figure 3:** (a) Kendall's Rank Correlation $\tau$ between daily maximum total sea level and daily maximum river discharge; and (b) number of joint occurrences per decade between extreme total sea levels and river discharge, at 0-day lag. Thick black lines in (a) represent that the dependence is statistically significant (95% confidence) at these sites. Note that the triangles, circles and squares show the river stations that discharge onto the west, east and south coasts, respectively.





**Figure 4:** (a) Kendall's Rank Correlation $\tau$ between daily maximum skew surge and daily maximum river discharge; and (b) number of joint occurrences per decade between extreme skew surge and extreme river discharge, at 0-day lag. Thick black lines in (a) represent that the dependence is statistically significant (95% confidence) at these sites. Note that the triangles, circles and squares show the river stations that discharge onto the west, east and south coasts, respectively.





**Figure 5:** Kendall's Rank Correlation $\tau$ plotted against day of lag at the follow sites: (a) Bournemouth; (b) Devonport; (c) Workington; (d) Ullapool; (e) Whitby; and (f) Cromer. The red dot shows the day with maximum lag.







**Figure 6:** (a) The lag day when the Kendall's Rank Correlation τ is maximum between daily maximum skew surge and daily maximum river discharge; and (b) when the lag day when the number of joint occurrences between high skew surge and high river discharge is maximum. Note that the triangles, circles and squares show the river stations that discharge onto the west, east and south coasts, respectively.





**Figure 7:** Meteorology conditions for Devonport: first column, sea level pressure (mbar); second column, wind speed (m/s) and direction (grey arrows); third column, precipitable water content (kg/m²); during (a, b and c) high skew surge events only, (d, e and f) both high skew surge and high river discharge events, and (g, h and i) extreme high river discharge events only. SD correspond to the averaged standard deviation over the grid for each variable across the selected events.





**Figure 8:** Meteorology conditions for Whitby: first column, sea level pressure (mbar); second column, wind speed (m/s) and direction (grey arrows); third column, precipitable water content (kg/m²); during (a, b and c) high skew surge only events, (d, e and f) both high skew surge and high river discharge events, and (g, h and i) extreme high river discharge only events. SD correspond to the averaged standard deviation over the grid for each variable across the selected events.





**Figure 9:** Storm tracks for Devonport (a, b and c) and Whitby (d, e and f) over Northern Europe. The first column (a, d) shows high skew surge only events. The second column (b, e) both high skew surge and high river discharge events. The third column (c, f) shows high river discharge only events. The blue line represents the mean storm track. Grey lines show individual storm tracks with the location of the storm at peak skew surge and/or peak river discharge shown by the red dot.



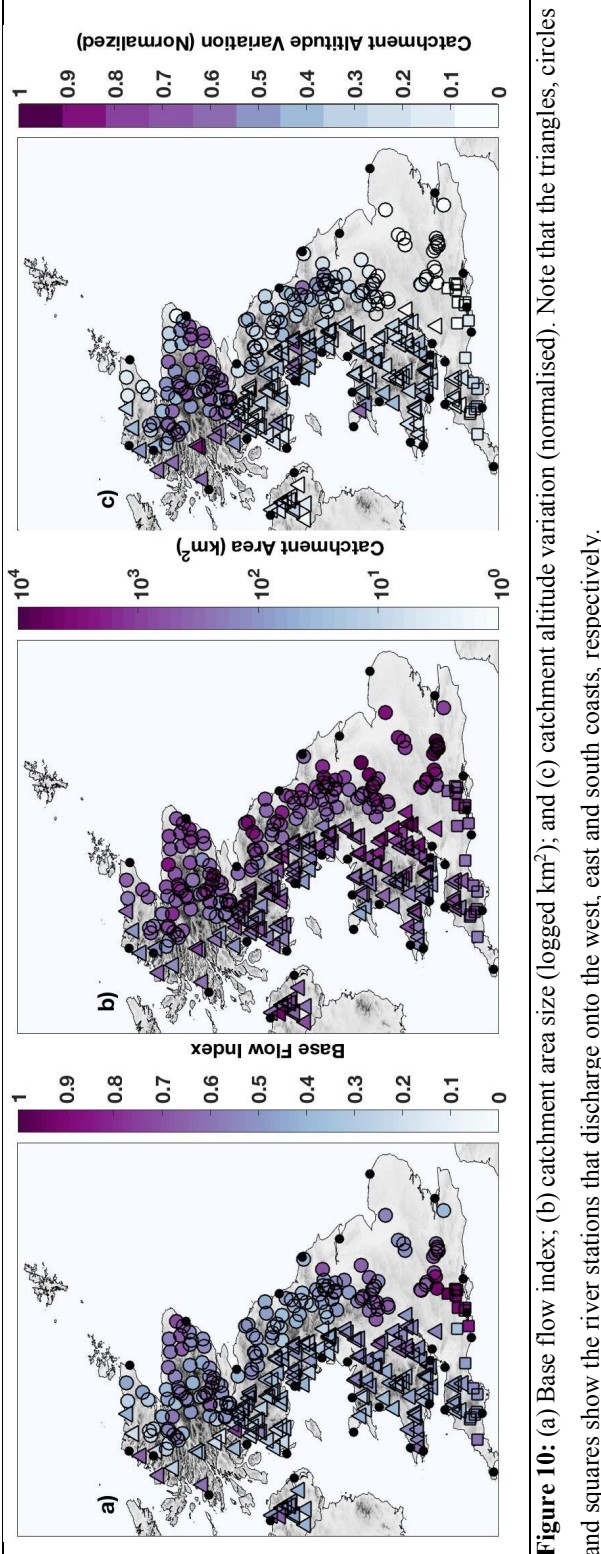

**Figure 10:** (a) Base flow index; (b) catchment area size (logged km²); and (c) catchment altitude variation (normalised). Note that the triangles, circles and squares show the river stations that discharge onto the west, east and south coasts, respectively.





**Figure 11:** Kendall's Rank Correlation τ between daily maximum skew surge and daily maximum river discharge with: (a) base flow index; (b) catchment area size (logged km²); (c) catchment altitude variation (normalised); and correlation of the day of lag with the largest Kendall's Rank Correlation τ with: (d) base flow index; (e) catchment area size (logged km²); (f) catchment altitude variation (normalised), for all sites.



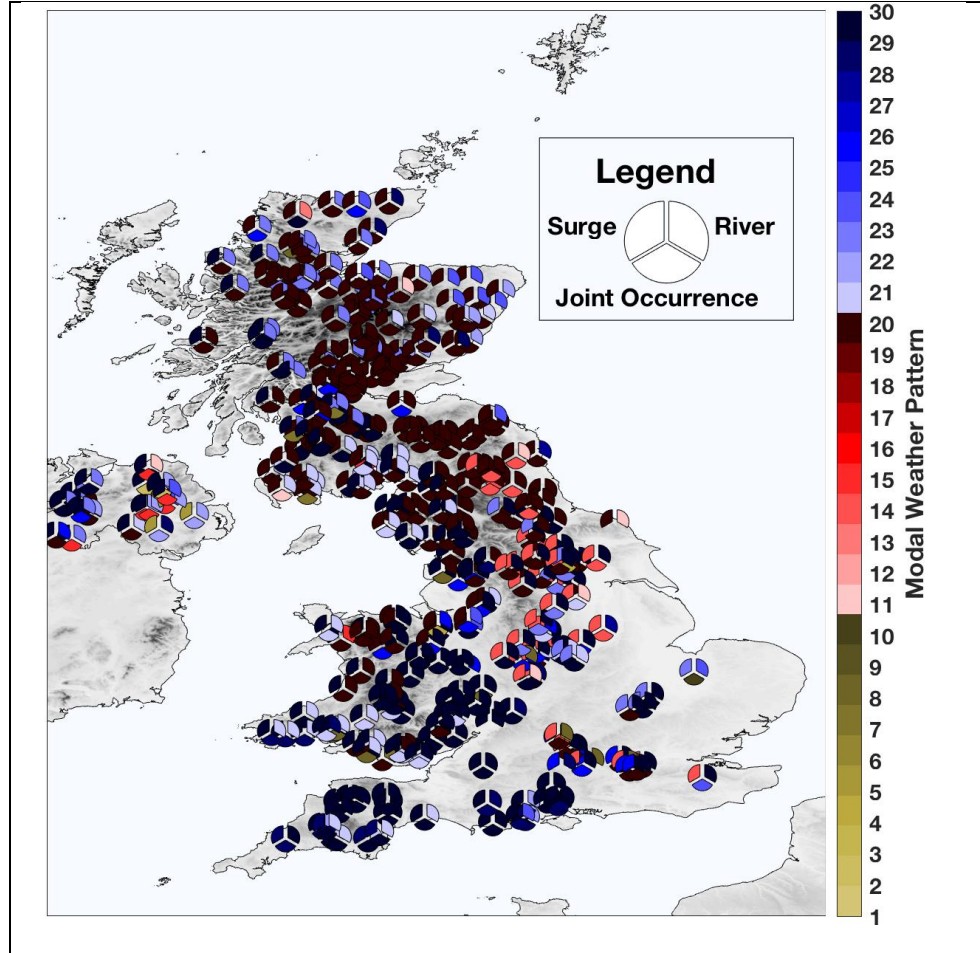

**Figure 12:** The modal weather pattern type (indicated by the colours in the legend) for extreme surge only events (top left segment), extreme river flow only events (top right segment), and extreme joint occurrence events (bottom segment) observed at the study locations.



## Tables

**Table 1**: The pairs of tide gauge sites and river discharge stations used in the study.

| Tide Gauge | Tide Gauge Latitude (deg) | Tide Gauge longitude (deg) | River Gauge ID | River | River Gauge Location | River Gauge Latitude (deg) | River Gauge longitude (deg) |
|---|---|---|---|---|---|---|---|
| Aberdeen | 57.14 | -2.08 | 12002 | Dee | Park | 57.08 | -2.33 |
| Aberdeen | 57.14 | -2.08 | 12001 | Dee | Woodend | 57.05 | -2.60 |
| Aberdeen | 57.14 | -2.08 | 12003 | Dee | Polhollick | 57.06 | -3.08 |
| Aberdeen | 57.14 | -2.08 | 11001 | Don | Parkhill | 57.22 | -2.19 |
| Aberdeen | 57.14 | -2.08 | 13007 | North Esk | Logie Mill | 56.77 | -2.49 |
| Aberdeen | 57.14 | -2.08 | 9002 | Deveron | Muiresk | 57.54 | -2.49 |
| Aberdeen | 57.14 | -2.08 | 11002 | Don | Haughton | 57.27 | -2.40 |
| Aberdeen | 57.14 | -2.08 | 13008 | South Esk | Brechin | 56.73 | -2.65 |
| Aberdeen | 57.14 | -2.08 | 12007 | Dee | Mar Lodge | 56.99 | -3.49 |
| Aberdeen | 57.14 | -2.08 | 11003 | Don | Bridge of Alford | 57.24 | -2.72 |
| Aberdeen | 57.14 | -2.08 | 9001 | Deveron | Avochie | 57.51 | -2.78 |
| Aberdeen | 57.14 | -2.08 | 10003 | Ythan | Ellon | 57.36 | -2.09 |
| Aberdeen | 57.14 | -2.08 | 12008 | Feugh | Heugh Head | 57.03 | -2.52 |
| Aberdeen | 57.14 | -2.08 | 13012 | South Esk | Gella Bridge | 56.78 | -3.03 |
| Avonmouth | 51.51 | -2.71 | 54057 | Severn | Haw Bridge | 51.95 | -2.23 |
| Avonmouth | 51.51 | -2.71 | 54032 | Severn | Saxons Lode | 52.05 | -2.20 |
| Avonmouth | 51.51 | -2.71 | 55023 | Wye | Redbrook | 51.80 | -2.69 |
| Avonmouth | 51.51 | -2.71 | 54001 | Severn | Bewdley | 52.38 | -2.32 |
| Avonmouth | 51.51 | -2.71 | 54095 | Severn | Buildwas | 52.64 | -2.52 |
| Avonmouth | 51.51 | -2.71 | 55002 | Wye | Belmont | 52.04 | -2.75 |
| Avonmouth | 51.51 | -2.71 | 54005 | Severn | Montford | 52.72 | -2.87 |
| Avonmouth | 51.51 | -2.71 | 55007 | Wye | Erwood | 52.09 | -3.35 |
| Avonmouth | 51.51 | -2.71 | 54028 | Vyrnwy | Llanymynech | 52.77 | -3.11 |
| Avonmouth | 51.51 | -2.71 | 53018 | Avon | Bathford | 51.40 | -2.31 |



| | | | | | | | |
|---|---|---|---|---|---|---|---|
| Avonmouth | 51.51 | -2.71 | 54029 | Teme | Knightsford Bridge | 52.20 | -2.39 |
| Avonmouth | 51.51 | -2.71 | 54002 | Avon | Evesham | 52.09 | -1.94 |
| Avonmouth | 51.51 | -2.71 | 54014 | Severn | Abermule | 52.55 | -3.23 |
| Avonmouth | 51.51 | -2.71 | 54008 | Teme | Tenbury | 52.31 | -2.59 |
| Avonmouth | 51.51 | -2.71 | 55003 | Lugg | Lugwardine | 52.06 | -2.66 |
| Avonmouth | 51.51 | -2.71 | 55012 | Irfon | Cilmery | 52.15 | -3.47 |
| Avonmouth | 51.51 | -2.71 | 55016 | Ithon | Disserth | 52.21 | -3.43 |
| Avonmouth | 51.51 | -2.71 | 54080 | Severn | Dolwen | 52.45 | -3.49 |
| Avonmouth | 51.51 | -2.71 | 54012 | Tern | Walcot | 52.71 | -2.60 |
| Avonmouth | 51.51 | -2.71 | 54038 | Tanat | Llanyblodwel | 52.79 | -3.11 |
| Avonmouth | 51.51 | -2.71 | 55026 | Wye | Ddol Farm | 52.30 | -3.50 |
| Avonmouth | 51.51 | -2.71 | 55029 | Monnow | Grosmont | 51.92 | -2.85 |
| Avonmouth | 51.51 | -2.71 | 55021 | Lugg | Butts Bridge | 52.23 | -2.73 |
| Avonmouth | 51.51 | -2.71 | 55032 | Elan | Caban Dam | 52.27 | -3.57 |
| Bangor | 54.66 | -5.67 | 205004 | Lagan | Newforge | 54.55 | -5.95 |
| Bangor | 54.66 | -5.67 | 203018 | Six-Mile Water | Antrim | 54.72 | -6.22 |
| Bangor | 54.66 | -5.67 | 203097 | Upper Bann | Moyallen | 54.39 | -6.39 |
| Barmouth | 52.72 | -4.05 | 64001 | Dyfi | Dyfi Bridge | 52.60 | -3.85 |
| Barmouth | 52.72 | -4.05 | 63001 | Ystwyth | Pont Llolwyn | 52.38 | -4.07 |
| Barmouth | 52.72 | -4.05 | 65001 | Glaslyn | Beddgelert | 53.01 | -4.10 |
| Bournemouth | 50.71 | -1.87 | 43021 | Avon | Knapp Mill | 50.75 | -1.78 |
| Bournemouth | 50.71 | -1.87 | 43003 | Avon | East Mills Total | 50.93 | -1.77 |
| Bournemouth | 50.71 | -1.87 | 43007 | Stour | Throop | 50.76 | -1.84 |
| Cromer | 52.93 | 1.30 | 33035 | Ely Ouse | Denver Complex | 52.58 | 0.35 |
| Cromer | 52.93 | 1.30 | 33026 | Bedford Ouse | Offord | 52.29 | -0.22 |
| Cromer | 52.93 | 1.30 | 33039 | Bedford Ouse | Roxton | 52.17 | -0.30 |





| Cromer | 52.93 | 1.30 | 33002 | Bedford Ouse | Bedford | 52.13 | -0.46 |
|---|---|---|---|---|---|---|---|
| Devonport | 50.37 | -4.19 | 47001 | Tamar | Gunnislake | 50.53 | -4.22 |
| Devonport | 50.37 | -4.19 | 46003 | Dart | Austins Bridge | 50.48 | -3.76 |
| Devonport | 50.37 | -4.19 | 47019 | Tamar | Polson Bridge | 50.64 | -4.33 |
| Devonport | 50.37 | -4.19 | 46002 | Teign | Preston | 50.56 | -3.62 |
| Devonport | 50.37 | -4.19 | 47015 | Tavy | Ludbrook | 50.49 | -4.15 |
| Devonport | 50.37 | -4.19 | 47006 | Lyd | Lifton Park | 50.64 | -4.28 |
| Fishguard | 52.01 | -4.98 | 62001 | Teifi | Glanteifi | 52.05 | -4.56 |
| Heysham | 54.03 | -2.92 | 72004 | Lune | Caton | 54.08 | -2.72 |
| Heysham | 54.03 | -2.92 | 71001 | Ribble | Samlesbury | 53.77 | -2.62 |
| Heysham | 54.03 | -2.92 | 71009 | Ribble | New Jumbles Rock | 53.83 | -2.45 |
| Heysham | 54.03 | -2.92 | 73010 | Leven | Newby Bridge | 54.27 | -2.97 |
| Heysham | 54.03 | -2.92 | 71006 | Ribble | Henthorn | 53.85 | -2.42 |
| Heysham | 54.03 | -2.92 | 72005 | Lune | Killington | 54.31 | -2.58 |
| Heysham | 54.03 | -2.92 | 73005 | Kent | Sedgwick | 54.28 | -2.75 |
| Heysham | 54.03 | -2.92 | 72011 | Rawthey | Brigflatts | 54.31 | -2.55 |
| Heysham | 54.03 | -2.92 | 71008 | Hodder | Hodder Place | 53.85 | -2.45 |
| Heysham | 54.03 | -2.92 | 71004 | Calder | Whalley Weir | 53.82 | -2.41 |
| Heysham | 54.03 | -2.92 | 71011 | Ribble | Arnford | 54.00 | -2.25 |
| Heysham | 54.03 | -2.92 | 72002 | Wyre | St Michaels | 53.86 | -2.82 |
| Heysham | 54.03 | -2.92 | 72015 | Lune | Lunes Bridge | 54.42 | -2.60 |
| Heysham | 54.03 | -2.92 | 74001 | Duddon | Duddon Hall | 54.30 | -3.24 |
| Hinkley | 51.22 | -3.13 | 45001 | Exe | Thorverton | 50.80 | -3.51 |
| Hinkley | 51.22 | -3.13 | 45002 | Exe | Stoodleigh | 50.95 | -3.51 |
| Hinkley | 51.22 | -3.13 | 45011 | Barle | Brushford | 51.02 | -3.53 |
| Holyhead | 53.31 | -4.62 | 65006 | Seiont | Peblig Mill | 53.14 | -4.25 |
| Ilfracombe | 51.21 | -4.11 | 50001 | Taw | Umberleigh | 50.99 | -3.99 |
| Ilfracombe | 51.21 | -4.11 | 50002 | Torridge | Torrington | 50.95 | -4.14 |





| Ilfracombe | 51.21 | -4.11 | 50006 | Mole | Woodleigh | 50.97 | -3.91 |
|---|---|---|---|---|---|---|---|
| Ilfracombe | 51.21 | -4.11 | 50010 | Torridge | Rockhay Bridge | 50.84 | -4.12 |
| Immingham | 53.63 | -0.19 | 28022 | Trent | North Muskham | 53.14 | -0.80 |
| Immingham | 53.63 | -0.19 | 28009 | Trent | Colwick | 52.95 | -1.08 |
| Immingham | 53.63 | -0.19 | 28007 | Trent | Shardlow | 52.86 | -1.33 |
| Immingham | 53.63 | -0.19 | 27009 | Ouse | Skelton | 53.99 | -1.13 |
| Immingham | 53.63 | -0.19 | 28019 | Trent | Drakelow Park | 52.78 | -1.65 |
| Immingham | 53.63 | -0.19 | 27003 | Aire | Beal Weir | 53.72 | -1.20 |
| Immingham | 53.63 | -0.19 | 27007 | Ure | Westwick Lock | 54.10 | -1.46 |
| Immingham | 53.63 | -0.19 | 27071 | Swale | Crakehill | 54.15 | -1.35 |
| Immingham | 53.63 | -0.19 | 27079 | Calder | Methley | 53.73 | -1.38 |
| Immingham | 53.63 | -0.19 | 28067 | Derwent | Church Wilne | 52.88 | -1.34 |
| Immingham | 53.63 | -0.19 | 27080 | Aire | Lemonroyd | 53.75 | -1.42 |
| Immingham | 53.63 | -0.19 | 27002 | Wharfe | Flint Mill Weir | 53.92 | -1.36 |
| Immingham | 53.63 | -0.19 | 27089 | Wharfe | Tadcaster | 53.89 | -1.27 |
| Immingham | 53.63 | -0.19 | 28085 | Derwent | St Mary's Bridge | 52.93 | -1.47 |
| Immingham | 53.63 | -0.19 | 27041 | Derwent | Buttercrambe | 54.02 | -0.88 |
| Immingham | 53.63 | -0.19 | 27034 | Derwent | Stamford Bridge | 54.27 | -1.71 |
| Immingham | 53.63 | -0.19 | 27021 | Don | Doncaster | 53.53 | -1.14 |
| Immingham | 53.63 | -0.19 | 27028 | Aire | Armley | 53.80 | -1.57 |
| Immingham | 53.63 | -0.19 | 28117 | Derwent | Whatstandwell | 53.09 | -1.51 |
| Immingham | 53.63 | -0.19 | 27043 | Wharfe | Addingham | 53.94 | -1.86 |
| Immingham | 53.63 | -0.19 | 28080 | Tame | Lea Marston Lakes | 52.54 | -1.69 |
| Immingham | 53.63 | -0.19 | 28018 | Dove | Marston on Dove | 52.86 | -1.65 |
| Immingham | 53.63 | -0.19 | 27090 | Swale | Catterick Bridge | 54.39 | -1.65 |





| | | | | | | | |
|---|---|---|---|---|---|---|---|
| Immingham | 53.63 | -0.19 | 28011 | Derwent | Matlock Bath | 53.12 | -1.56 |
| Immingham | 53.63 | -0.19 | 28012 | Trent | Yoxall | 52.76 | -1.80 |
| Immingham | 53.63 | -0.19 | 28074 | Soar | Kegworth | 52.83 | -1.27 |
| Immingham | 53.63 | -0.19 | 28093 | Soar | Pillings Lock | 52.76 | -1.16 |
| Immingham | 53.63 | -0.19 | 27062 | Nidd | Skip Bridge | 54.00 | -1.26 |
| Immingham | 53.63 | -0.19 | 27029 | Calder | Elland | 53.69 | -1.81 |
| Immingham | 53.63 | -0.19 | 27001 | Nidd | Hunsingore Weir | 53.97 | -1.35 |
| Immingham | 53.63 | -0.19 | 28008 | Dove | Rocester Weir | 52.95 | -1.83 |
| Immingham | 53.63 | -0.19 | 27035 | Aire | Kildwick Bridge | 53.91 | -1.98 |
| Immingham | 53.63 | -0.19 | 28043 | Derwent | Chatsworth | 53.21 | -1.61 |
| Immingham | 53.63 | -0.19 | 28014 | Sow | Milford | 52.79 | -2.04 |
| Immingham | 53.63 | -0.19 | 28003 | Tame | Water Orton | 52.52 | -1.75 |
| Immingham | 53.63 | -0.19 | 27006 | Don | Hadfields Weir | 53.41 | -1.41 |
| Immingham | 53.63 | -0.19 | 27053 | Nidd | Birstwith | 54.04 | -1.65 |
| Kinlochbervie | 58.46 | -5.05 | 96002 | Naver | Apigill | 58.48 | -4.21 |
| Kinlochbervie | 58.46 | -5.05 | 96004 | Strathmore | Allnabad | 58.35 | -4.65 |
| Leith | 55.99 | -3.18 | 15006 | Tay | Ballathie | 56.51 | -3.39 |
| Leith | 55.99 | -3.18 | 15003 | Tay | Caputh | 56.54 | -3.49 |
| Leith | 55.99 | -3.18 | 21009 | Tweed | Norham | 55.72 | -2.16 |
| Leith | 55.99 | -3.18 | 15012 | Tummel | Pitlochry | 56.70 | -3.72 |
| Leith | 55.99 | -3.18 | 21021 | Tweed | Sprouston | 55.61 | -2.39 |
| Leith | 55.99 | -3.18 | 15007 | Tay | Pitnacree | 56.66 | -3.76 |
| Leith | 55.99 | -3.18 | 15016 | Tay | Kenmore | 56.60 | -3.99 |
| Leith | 55.99 | -3.18 | 18011 | Forth | Craigforth | 56.14 | -3.97 |
| leith | 55.99 | -3.18 | 21006 | Tweed | Boleside | 55.59 | -2.80 |
| Leith | 55.99 | -3.18 | 16004 | Earn | Forteviot Bridge | 56.35 | -3.55 |
| Leith | 55.99 | -3.18 | 18003 | Teith | Bridge of Teith | 56.19 | -4.06 |





| Leith | 55.99 | -3.18 | 16001 | Earn | Kinkell Bridge | 56.33 | -3.73 |
|-------|-------|-------|-------|------|----------------|-------|-------|
| Leith | 55.99 | -3.18 | 21008 | Teviot | Ormiston Mill | 55.55 | -2.47 |
| Leith | 55.99 | -3.18 | 15034 | Garry | Killiecrankie | 56.75 | -3.80 |
| Leith | 55.99 | -3.18 | 21003 | Tweed | Peebles | 55.65 | -3.18 |
| Leith | 55.99 | -3.18 | 15024 | Dochart | Killin | 56.46 | -4.33 |
| Leith | 55.99 | -3.18 | 18010 | Forth | Gargunnock | 56.13 | -4.07 |
| Leith | 55.99 | -3.18 | 21007 | Ettrick Water | Lindean | 55.57 | -2.82 |
| Leith | 55.99 | -3.18 | 15025 | Ericht | Craighall | 56.61 | -3.35 |
| Leith | 55.99 | -3.18 | 18008 | Leny | Anie | 56.26 | -4.29 |
| Leith | 55.99 | -3.18 | 15011 | Lyon | Comrie Bridge | 56.61 | -3.98 |
| Leith | 55.99 | -3.18 | 21005 | Tweed | Lyne Ford | 55.64 | -3.26 |
| Leith | 55.99 | -3.18 | 21012 | Teviot | Hawick | 55.43 | -2.76 |
| Leith | 55.99 | -3.18 | 15010 | Isla | Wester Cardean | 56.61 | -3.15 |
| Leith | 55.99 | -3.18 | 18015 | Eas Gobhain | Loch Venachar | 56.23 | -4.26 |
| Leith | 55.99 | -3.18 | 15039 | Tilt | Marble Lodge | 56.82 | -3.82 |
| Leith | 55.99 | -3.18 | 15023 | Braan | Hermitage | 56.56 | -3.61 |
| Leith | 55.99 | -3.18 | 18005 | Allan Water | Bridge of Allan | 56.16 | -3.96 |
| Leith | 55.99 | -3.18 | 15041 | Lyon | Camusvrachan | 56.60 | -4.25 |
| Leith | 55.99 | -3.18 | 21022 | Whiteadder Water | Hutton Castle | 55.79 | -2.19 |
| Leith | 55.99 | -3.18 | 21011 | Yarrow Water | Philiphaugh | 55.54 | -2.89 |
| Leith | 55.99 | -3.18 | 19001 | Almond | Craigiehall | 55.96 | -3.34 |
| Leith | 55.99 | -3.18 | 15013 | Almond | Almondbank | 56.42 | -3.51 |
| Leith | 55.99 | -3.18 | 18001 | Allan Water | Kinbuck | 56.23 | -3.95 |





| | | | | | | | |
|---|---|---|---|---|---|---|---|
| Leith | 55.99 | -3.18 | 16003 | Ruchill Water | Cultybraggan | 56.36 | -4.00 |
| Leith | 55.99 | -3.18 | 21020 | Yarrow Water | Gordon Arms | 55.51 | -3.09 |
| Liverpool | 53.45 | -3.02 | 67027 | Dee | Ironbridge | 53.13 | -2.87 |
| Liverpool | 53.45 | -3.02 | 67033 | Dee | Chester Suspension Bridge | 53.19 | -2.88 |
| Liverpool | 53.45 | -3.02 | 67015 | Dee | Manley Hall | 52.97 | -2.97 |
| Liverpool | 53.45 | -3.02 | 69002 | Irwell | Adelphi Weir | 53.49 | -2.26 |
| Liverpool | 53.45 | -3.02 | 67001 | Dee | Bala | 52.91 | -3.58 |
| Liverpool | 53.45 | -3.02 | 69007 | Mersey | Ashton Weir | 53.44 | -2.34 |
| Liverpool | 53.45 | -3.02 | 68001 | Weaver | Ashbrook | 53.17 | -2.49 |
| Liverpool | 53.45 | -3.02 | 67006 | Alwen | Druid | 52.98 | -3.43 |
| Liverpool | 53.45 | -3.02 | 68003 | Dane | Rudheath | 53.24 | -2.50 |
| Llandudno | 53.33 | -3.83 | 66011 | Conwy | Cwmlanerch | 53.11 | -3.79 |
| Llandudno | 53.33 | -3.83 | 66025 | Clwyd | Pont Dafydd | 53.26 | -3.43 |
| Llandudno | 53.33 | -3.83 | 66001 | Clwyd | Pont-y-Cambwll | 53.23 | -3.39 |
| Llandudno | 53.33 | -3.83 | 66012 | Lledr | Pont Gethin | 53.07 | -3.81 |
| Milford Haven | 51.71 | -5.05 | 60003 | Taf | Clog-y-Fran | 51.81 | -4.56 |
| Milford Haven | 51.71 | -5.05 | 61002 | Eastern Cleddau | Canaston Bridge | 51.80 | -4.80 |
| Milford Haven | 51.71 | -5.05 | 61001 | Western Cleddau | Prendergast Mill | 51.82 | -4.97 |
| Millport | 55.75 | -4.91 | 84013 | Clyde | Daldowie | 55.83 | -4.12 |
| Millport | 55.75 | -4.91 | 85001 | Leven | Linnbrane | 55.99 | -4.58 |
| Millport | 55.75 | -4.91 | 84005 | Clyde | Blairston | 55.80 | -4.07 |
| Millport | 55.75 | -4.91 | 84003 | Clyde | Hazelbank | 55.69 | -3.85 |
| Millport | 55.75 | -4.91 | 84018 | Clyde | Tulliford Mill | 55.64 | -3.76 |
| Millport | 55.75 | -4.91 | 89003 | Orchy | Glen Orchy | 56.45 | -4.86 |





| | | | | | | | |
|---|---|---|---|---|---|---|---|
| Millport | 55.75 | -4.91 | 84004 | Clyde | Sills of Clyde | 55.66 | -3.70 |
| Millport | 55.75 | -4.91 | 83006 | Ayr | Mainholm | 55.46 | -4.59 |
| Millport | 55.75 | -4.91 | 86002 | Eachaig | Eckford | 56.02 | -4.99 |
| Millport | 55.75 | -4.91 | 83005 | Irvine | Shewalton | 55.60 | -4.63 |
| Millport | 55.75 | -4.91 | 84001 | Kelvin | Killermont | 55.91 | -4.31 |
| Millport | 55.75 | -4.91 | 84014 | Avon Water | Fairholm | 55.74 | -3.99 |
| Millport | 55.75 | -4.91 | 85002 | Endrick Water | Gaidrew | 56.05 | -4.44 |
| Millport | 55.75 | -4.91 | 82002 | Doon | Auchendrane | 55.41 | -4.63 |
| Millport | 55.75 | -4.91 | 84015 | Kelvin | Dryfield | 55.94 | -4.18 |
| Millport | 55.75 | -4.91 | 82001 | Girvan | Robstone | 55.26 | -4.81 |
| Millport | 55.75 | -4.91 | 84012 | White Cart Water | Hawkhead | 55.84 | -4.40 |
| Millport | 55.75 | -4.91 | 83009 | Garnock | Kilwinning | 55.65 | -4.69 |
| Millport | 55.75 | -4.91 | 83013 | Irvine | Glenfield | 55.60 | -4.49 |
| Millport | 55.75 | -4.91 | 85003 | Falloch | Glen Falloch | 56.34 | -4.72 |
| Millport | 55.75 | -4.91 | 83004 | Lugar Water | Langholm | 55.47 | -4.36 |
| Millport | 55.75 | -4.91 | 83003 | Ayr | Catrine | 55.50 | -4.34 |
| Mumbles | 51.57 | -3.98 | 60010 | Tywi | Capel Dewi | 51.86 | -4.20 |
| Mumbles | 51.57 | -3.98 | 59001 | Tawe | Ynystanglws | 51.68 | -3.90 |
| Mumbles | 51.57 | -3.98 | 60002 | Cothi | Felin Mynachdy | 51.88 | -4.17 |
| Mumbles | 51.57 | -3.98 | 60007 | Tywi | Dolau Hirion | 52.01 | -3.81 |
| Mumbles | 51.57 | -3.98 | 58002 | Neath | Resolven | 51.70 | -3.72 |
| Mumbles | 51.57 | -3.98 | 58001 | Ogmore | Bridgend | 51.50 | -3.58 |
| Mumbles | 51.57 | -3.98 | 58012 | Afan | Marcroft Weir | 51.60 | -3.78 |
| Mumbles | 51.57 | -3.98 | 60006 | Gwili | Glangwili | 51.87 | -4.28 |
| Newlyn | 50.10 | -5.54 | 49001 | Camel | Denby | 50.48 | -4.80 |



| Newport | 51.55 | -2.99 | 56001 | Usk | Chainbridge | 51.74 | -2.95 |
|---|---|---|---|---|---|---|---|
| Newport | 55.01 | -1.44 | 23003 | North Tyne | Reaverhill | 55.05 | -2.15 |
| Newport | 51.55 | -2.99 | 57005 | Taff | Pontypridd | 51.60 | -3.33 |
| Newport | 51.55 | -2.99 | 56002 | Ebbw | Rhiwderin | 51.59 | -3.07 |
| Newport | 51.55 | -2.99 | 57007 | Taff | Fiddlers Elbow | 51.65 | -3.32 |
| Newport | 51.55 | -2.99 | 57006 | Rhondda | Trehafod | 51.61 | -3.37 |
| Newport | 51.55 | -2.99 | 57008 | Rhymney | Llanedeyrn | 51.53 | -3.12 |
| North Shields | 55.01 | -1.44 | 23001 | Tyne | Bywell | 54.95 | -1.94 |
| North Shields | 55.01 | -1.44 | 23004 | South Tyne | Haydon Bridge | 54.98 | -2.22 |
| North Shields | 55.01 | -1.44 | 24009 | Wear | Chester le Street | 54.85 | -1.56 |
| North Shields | 55.01 | -1.44 | 24001 | Wear | Sunderland Bridge | 54.73 | -1.59 |
| North Shields | 55.01 | -1.44 | 23006 | South Tyne | Featherstone | 54.94 | -2.51 |
| North Shields | 55.01 | -1.44 | 22001 | Coquet | Morwick | 55.33 | -1.63 |
| North Shields | 55.01 | -1.44 | 23022 | North Tyne | Uglydub | 55.18 | -2.45 |
| North Shields | 55.01 | -1.44 | 23005 | North Tyne | Tarset | 55.17 | -2.35 |
| North Shields | 55.01 | -1.44 | 24008 | Wear | Witton Park | 54.67 | -1.73 |
| North Shields | 55.01 | -1.44 | 23008 | Rede | Rede Bridge | 55.14 | -2.21 |
| North Shields | 55.01 | -1.44 | 22009 | Coquet | Rothbury | 55.31 | -1.89 |
| Portpatrick | 54.84 | -5.12 | 81002 | Cree | Newton Stewart | 54.96 | -4.48 |
| Portpatrick | 54.84 | -5.12 | 81004 | Bladnoch | Low Malzie | 54.86 | -4.52 |
| Portpatrick | 54.84 | -5.12 | 81006 | Water of Minnoch | Minnoch Bridge | 55.04 | -4.57 |





| | | | | | | | |
|---|---|---|---|---|---|---|---|
| Portpatrick | 54.84 | -5.12 | 81003 | Luce | Airyhemming | 54.90 | -4.84 |
| Portrush | 55.21 | -6.66 | 203040 | Lower Bann | Movanagher | 54.98 | -6.55 |
| Portrush | 55.21 | -6.66 | 201010 | Mourne | Drumnabuoy House | 54.81 | -7.46 |
| Portrush | 55.21 | -6.66 | 203093 | Main | Shane's Viaduct | 54.74 | -6.31 |
| Portrush | 55.21 | -6.66 | 203010 | Blackwater | Maydown Bridge | 54.41 | -6.74 |
| Portrush | 55.21 | -6.66 | 201009 | Owenkillew | Crosh | 54.73 | -7.35 |
| Portrush | 55.21 | -6.66 | 201008 | Derg | Castlederg | 54.71 | -7.59 |
| Portrush | 54.84 | -5.12 | 82003 | Stinchar | Balnowlart | 55.11 | -4.97 |
| Portrush | 55.21 | -6.66 | 203012 | Ballinderry | Ballinderry Bridge | 54.66 | -6.56 |
| Portrush | 55.21 | -6.66 | 203020 | Moyola | Moyola New Bridge | 54.66 | -6.52 |
| Portrush | 55.21 | -6.66 | 201006 | Drumragh | Campsie Bridge | 54.60 | -7.29 |
| Portrush | 55.21 | -6.66 | 236005 | Colebrooke | Ballindarragh Bridge | 54.27 | -7.49 |
| Portrush | 55.21 | -6.66 | 202002 | Faughan | Drumahoe | 54.98 | -7.28 |
| Portrush | 55.21 | -6.66 | 204001 | Bush | Seneirl Bridge | 55.16 | -6.52 |
| Portrush | 55.21 | -6.66 | 201005 | Camowen | Camowen Terrace | 54.60 | -7.29 |
| Portrush | 55.21 | -6.66 | 203011 | Main | Dromona | 54.92 | -6.37 |
| Portrush | 55.21 | -6.66 | 203092 | Main | Dunminning | 54.94 | -6.36 |
| Portrush | 55.21 | -6.66 | 236007 | Sillees | Drumrainey Bridge | 54.31 | -7.69 |
| Portrush | 55.21 | -6.66 | 201002 | Fairywater | Dudgeon Bridge | 54.63 | -7.37 |
| Portrush | 55.21 | -6.66 | 203027 | Braid | Ballee | 54.85 | -6.29 |
| Portsmouth | 50.80 | -1.11 | 42004 | Test | Broadlands | 50.97 | -1.50 |





| | | | | | | | |
|---|---|---|---|---|---|---|---|
| Portsmouth | 50.80 | -1.11 | 42023 | Itchen | Riverside Park | 50.94 | -1.37 |
| Portsmouth | 50.80 | -1.11 | 42024 | Test | Chilbolton Total | 51.15 | -1.45 |
| Portsmouth | 50.80 | -1.11 | 42010 | Itchen | Highbridge & Allbrook Total | 50.99 | -1.34 |
| Sheerness | 51.45 | 0.74 | 39001 | Thames | Kingston | 51.41 | -0.31 |
| Sheerness | 51.45 | 0.74 | 39072 | Thames | Royal Windsor Park | 51.49 | -0.59 |
| Sheerness | 51.45 | 0.74 | 39121 | Thames | Walton | 51.39 | -0.42 |
| Sheerness | 51.45 | 0.74 | 39111 | Thames | Staines | 51.43 | -0.51 |
| Sheerness | 51.45 | 0.74 | 39130 | Thames | Reading | 51.46 | -0.97 |
| Sheerness | 51.45 | 0.74 | 39002 | Thames | Days Weir | 51.64 | -1.18 |
| Sheerness | 51.45 | 0.74 | 39046 | Thames | Sutton Courtenay | 51.65 | -1.25 |
| Sheerness | 51.45 | 0.74 | 39129 | Thames | Farmoor | 51.76 | -1.36 |
| Sheerness | 51.45 | 0.74 | 39008 | Thames | Eynsham | 51.77 | -1.35 |
| Sheerness | 51.45 | 0.74 | 40003 | Medway | Teston / East Farleigh | 51.25 | 0.45 |
| Sheerness | 51.45 | 0.74 | 39016 | Kennet | Theale | 51.43 | -1.07 |
| Sheerness | 51.45 | 0.74 | 39079 | Wey | Weybridge | 51.37 | -0.46 |
| Sheerness | 51.45 | 0.74 | 39104 | Mole | Esher | 51.38 | -0.37 |
| Sheerness | 51.45 | 0.74 | 39103 | Kennet | Newbury | 51.40 | -1.32 |
| Tobermory | 56.62 | -6.06 | 91002 | Lochy | Camisky | 56.88 | -5.05 |
| Tobermory | 56.62 | -6.06 | 92001 | Shiel | Shielfoot | 56.76 | -5.83 |
| Tobermory | 56.62 | -6.06 | 90003 | Nevis | Claggan | 56.82 | -5.09 |
| Ullapool | 57.90 | -5.16 | 94001 | Ewe | Poolewe | 57.76 | -5.61 |
| Ullapool | 57.90 | -5.16 | 93001 | Carron | New Kelso | 57.43 | -5.44 |
| Ullapool | 57.90 | -5.16 | 95001 | Inver | Little Assynt | 58.17 | -5.16 |
| Ullapool | 57.90 | -5.16 | 95002 | Broom | Inverbroom | 57.81 | -5.06 |





| | | | | | | | |
|---|---|---|---|---|---|---|---|
| Weymouth | 50.61 | -2.45 | 44001 | Frome | East Stoke Total | 50.68 | -2.19 |
| Weymouth | 50.61 | -2.45 | 43009 | Stour | Hammoon | 50.93 | -2.26 |
| Weymouth | 50.61 | -2.45 | 45004 | Axe | Whitford | 50.75 | -3.05 |
| Whitby | 54.49 | -0.61 | 25009 | Tees | Low Moor | 54.49 | -1.44 |
| Whitby | 54.49 | -0.61 | 25001 | Tees | Broken Scar | 54.52 | -1.60 |
| Whitby | 54.49 | -0.61 | 25008 | Tees | Barnard Castle | 54.54 | -1.93 |
| Whitby | 54.49 | -0.61 | 25018 | Tees | Middleton in Teesdale | 54.62 | -2.08 |
| Whitby | 54.49 | -0.61 | 27092 | Esk | Briggswath | 54.46 | -0.65 |
| Wick | 58.44 | -3.09 | 6007 | Ness | Ness-side | 57.45 | -4.26 |
| Wick | 58.44 | -3.09 | 8006 | Spey | Boat o Brig | 57.55 | -3.14 |
| Wick | 58.44 | -3.09 | 4001 | Conon | Moy Bridge | 57.56 | -4.54 |
| Wick | 58.44 | -3.09 | 8010 | Spey | Grantown | 57.32 | -3.61 |
| Wick | 58.44 | -3.09 | 5003 | Glass | Kerrow Wood | 57.35 | -4.74 |
| Wick | 58.44 | -3.09 | 8005 | Spey | Boat of Garten | 57.25 | -3.75 |
| Wick | 58.44 | -3.09 | 8002 | Spey | Kinrara | 57.15 | -3.85 |
| Wick | 58.44 | -3.09 | 7002 | Findhorn | Forres | 57.61 | -3.64 |
| Wick | 58.44 | -3.09 | 5002 | Farrar | Struy | 57.43 | -4.68 |
| Wick | 58.44 | -3.09 | 3003 | Oykel | Easter Turnaig | 57.96 | -4.70 |
| Wick | 58.44 | -3.09 | 8004 | Avon | Delnashaugh | 57.40 | -3.36 |
| Wick | 58.44 | -3.09 | 7001 | Findhorn | Shenachie | 57.38 | -3.95 |
| Wick | 58.44 | -3.09 | 2001 | Helmsdale | Kilphedir | 58.14 | -3.70 |
| Wick | 58.44 | -3.09 | 2002 | Brora | Bruachrobie | 58.01 | -3.88 |
| Wick | 58.44 | -3.09 | 3002 | Carron | Sgodachail | 57.89 | -4.55 |
| Wick | 58.44 | -3.09 | 97002 | Thurso | Halkirk | 58.52 | -3.49 |
| Wick | 58.44 | -3.09 | 8013 | Feshie | Feshie Bridge | 57.12 | -3.90 |
| Wick | 58.44 | -3.09 | 3004 | Cassley | Rosehall | 57.98 | -4.59 |
| Wick | 58.44 | -3.09 | 4005 | Meig | Glenmeanie | 57.53 | -4.87 |
| Wick | 58.44 | -3.09 | 4003 | Alness | Alness | 57.70 | -4.26 |





| | | | | | | | |
|---|---|---|---|---|---|---|---|
| Wick | 58.44 | -3.09 | 8007 | Spey | Invertruim | 57.04 | -4.16 |
| Wick | 58.44 | -3.09 | 8009 | Dulnain | Balnaan Bridge | 57.30 | -3.70 |
| Wick | 58.44 | -3.09 | 4004 | Blackwater | Contin | 57.57 | -4.59 |
| Wick | 58.44 | -3.09 | 6009 | Moriston | Levishie | 57.22 | -4.65 |
| Wick | 58.44 | -3.09 | 7004 | Nairn | Firhall | 57.57 | -3.87 |
| Wick | 58.44 | -3.09 | 5004 | Glass | Fasnakyle | 57.32 | -4.80 |
| Wick | 58.44 | -3.09 | 96001 | Halladale | Halladale | 58.48 | -3.90 |
| Workington | 54.65 | -3.57 | 76007 | Eden | Sheepmount | 54.90 | -2.95 |
| Workington | 54.65 | -3.57 | 80002 | Dee | Glenlochar | 54.96 | -3.98 |
| Workington | 54.65 | -3.57 | 78003 | Annan | Brydekirk | 55.02 | -3.27 |
| Workington | 54.65 | -3.57 | 79002 | Nith | Friars Carse | 55.15 | -3.69 |
| Workington | 54.65 | -3.57 | 75002 | Derwent | Camerton | 54.66 | -3.49 |
| Workington | 54.65 | -3.57 | 77002 | Esk | Canonbie | 55.07 | -2.94 |
| Workington | 54.65 | -3.57 | 79006 | Nith | Drumlanrig | 55.27 | -3.80 |
| Workington | 54.65 | -3.57 | 75003 | Derwent | Ouse Bridge | 54.68 | -3.24 |
| Workington | 54.65 | -3.57 | 76003 | Eamont | Udford | 54.67 | -2.66 |
| Workington | 54.65 | -3.57 | 76005 | Eden | Temple Sowerby | 54.65 | -2.61 |
| Workington | 54.65 | -3.57 | 75005 | Derwent | Portinscale | 54.60 | -3.16 |
| Workington | 54.65 | -3.57 | 77003 | Liddel Water | Rowanburnfoot | 55.07 | -2.92 |
| Workington | 54.65 | -3.57 | 78006 | Annan | Woodfoot | 55.29 | -3.42 |
| Workington | 54.65 | -3.57 | 79005 | Cluden Water | Fiddlers Ford | 55.10 | -3.68 |
| Workington | 54.65 | -3.57 | 76015 | Eamont | Pooley Bridge | 54.62 | -2.82 |
| Workington | 54.65 | -3.57 | 78005 | Kinnel Water | Bridgemuir | 55.15 | -3.43 |
| Workington | 54.65 | -3.57 | 76008 | Irthing | Greenholme | 54.91 | -2.80 |
| Workington | 58.44 | -3.09 | 4006 | Bran | Dosmucheran | 57.60 | -5.01 |
| Workington | 54.65 | -3.57 | 80001 | Urr | Dalbeattie | 54.93 | -3.84 |
| Workington | 54.65 | -3.57 | 79003 | Nith | Hall Bridge | 55.39 | -4.08 |





| Workington | 54.65 | -3.57 | 79004 | Scar Water | Capenoch | 55.23 | -3.82 |
| Workington | 54.65 | -3.57 | 75004 | Cocker | Southwaite Bridge | 54.64 | -3.35 |
| Workington | 54.65 | -3.57 | 74005 | Ehen | Braystones | 54.44 | -3.53 |

**Table 2**: Correlation between catchment variables and: (i) the number of joint occurrences per decade between high skew surges and river discharge; and (ii) the lag day when there is the maximum number of joint occurrences between high skew surge and high river discharge. Bold text indicates statistical significance at a 95% confidence interval.

| Catchment Variable | All Sites | | Coastal Sites | |
| | Dependence | Lag | Dependence | Lag |
| --- | --- | --- | --- | --- |
| BFI | **-0.50** | **0.21** | **-0.48** | 0.17 |
| Catchment Area Size | **-0.31** | **0.12** | **-0.33** | 0.13 |
| Altitude Variation | **0.16** | -0.032 | **0.34** | 0.17 |