# Peer review of "Assessing the characteristics and drivers of compound flooding events around the UK coast"

_Hydrology and Earth System Sciences, 2018_

## Referee Comment (RC1) · Michalis Vousdoukas (Referee) · 1 Mar 2019

Michalis Vousdoukas (Referee)

michail.vousdoukas@ec.europa.eu

I found the paper very well written and rather complete, so I would like to congratulate the authors for their efforts and add very few comments.

Page 7 line 19: 2 degrees is rather coarse resolution. It makes me skeptical whether it would make sense to opt for a shorter time horizon of the analysis, in order to benefit from a finer reanalysis. Of course I am not suggesting that as I think that the meteorological analysis is not the main part of the study, but at least the authors should discuss this limitation. Ideally they could present some comparisons with results using ERA-5 or ERA-INTERIM for some points with data only for the more recent years.

[Figure]

There is a lot of space for improvement regarding the figures. Before publication the authors should try to improve by removing the tables, label subplots with a, b, c...

The literature review could be updated further, for example paragraph one of the introduction omits several recent papers beyond 2017

---

## Referee Comment (RC2) · Daniel Bachmann (Referee) · 4 Mar 2019

Dear authors, thanks for the very interesting paper. In the following you will find my comments, suggestions and opinion to your contribution.

**Summary**

The paper "Assessing the characteristics and drivers of compound flooding events around the UK coast" is about a statistical analysis of compound flooding events (mainly storm surges and river flooding) for the UK. A broad data-set about high storm surge water levels and river discharges are statistically analysed. Additionally, meteo-

rological data and data about the river catchment are included into the analysis. Three main hypotheses are tested:

-Map the spatial dependency between storm surges and high river discharges

-Map meteorological conditions which drive compound / non-compound events

-Map dependency of catchment characteristics and compound / non-compound events

All three hypotheses are well confirmed by the results of the statistical analyses.

Focusing on hydrological and storm surge events the contribution addresses a relevant topic, which matches the scope of the envisaged journal HESS very well (e.g. study of the spatial and temporal characteristics of the global water resources).

**General remarks/suggestions to style and language**

-Note: I am not capable for any review about the English language because I am not a native speaker. For me, the text is comprehensible.

-Sometimes it is confusing, which figure is relevant, which is for the supplement etc. Maybe check if really all tables, figures etc. are required

**General remarks to content**

-A lot of the statements requires very good knowledge of the British geography, e.g. "...closest to the tide gauges of Bournemouth, Devonport, Workington, Ullapool, Whitby and Cromer". I excuse my ignorance of the British geography, but I cannot reproduce some of the given statements with help of the figures. I have also no perfect solution, just proposals: Give numbers to (at least) the coastal gauge stations; make detailed figures for some statements; try to focus on special stations etc.

-I missed a bit the focus to one important questions, which arise from the risk analysis / management point of view: does these compound events produce additional damages, because of their compound occurrence? It is stated in the introduction: e.g.

[Figure]

"Zscheischler et al. (2018) define compound events as the combination of multiple drivers and/or hazards that contributes to societal or environmental risk." Additionally, some examples are given from the US and the Adriatic coast. But I miss a bit more focus on the UK coast and the events which were analysed in the following chapters. In the discussion section this question is partly discussed; It comes very suddenly and unexpectedly. I suggest to make it much more prominent at the beginning of the paper (at least some remarks in the introduction). Do we observe additional damages due to compound events at the UK coast? A second question is, why "just extreme events" were analysed. It is also stated in the introduction: "combinations of events that are not themselves extremes but lead to an extreme event when combined"! I think especially these events are the important events which we've miss until now in risk analysis. Maybe make a remark that this point is excluded from the analysis (if it is not done somewhere).

-I would like to note a very similar remark about the discussion of the forecasting strategies also in the "discussion"-chapter. It comes also very suddenly and unexpectedly in this chapter. I think it could be also (at least) mentioned in the introduction.

-Sometimes I would prefer that the hypotheses are more feed with physical reasoning, especially when the results are described: in some cases, it is done, in other cases it is done in the "discussion"-chapter, sometimes it could be a bit more pronounced. The question is: just give the physical explanation in the "discussion"-chapter or immediately after the description of the results? In this case, where a lot of results are described a direct explanation could be preferable (suggestions!). If the explanations first come in the discussion-chapter, some important details are already forgotten by the reader. Moreover, the purpose of a discussion chapter is more (as far as I know): discuss your findings critically, what are the limitations?, what could be better?, in which direction will it go? Etc.

-Finally, I have a bit the feeling, that too much information (esp. results) is given to the reader. Therefore, much more detailed descriptions are sometimes lacking. Maybe a
review of important information (text / figures etc.) should be done.

**Detailed Comments and proposals**

See attached document

**Conclusion of the review**

The presented article is a very interesting contribution about the statistical analysis of compound flood events. I am convinced that this paper is a valuable contribution to the journal HESS. I hope my suggestions, remarks, comments can help to improve this manuscript further.

Please also note the supplement to this comment:
https://www.hydrol-earth-syst-sci-discuss.net/hess-2018-632/hess-2018-632-RC2-supplement.pdf

---

## Author Comment (AC1) · 29 Apr 2019

Dear Editor,

We thank the reviewers for their thoughtful and useful comments. We are pleased that the reviewers find the topic to be relevant and interesting. The reviewers have raised a number of important comments, which we address in this rebuttal and follow up in the revised manuscript. Below we have listed each of their comments, and then in red text, outline how we have addressed each comment in turn.

Regards,

The Authors

**Reviewer Comments**

Reviewer 1 (Michalis Vousdoukas)

I found the paper very well written and rather complete, so I would like to congratulate the authors for their efforts and add very few comments. Page 7 line 19: 2 degrees is rather coarse resolution. It makes me skeptical whether it would make sense to opt for a shorter time horizon of the analysis, in order to benefit from a finer reanalysis. Of course I am not suggesting that as I think that the meteorological analysis is not the main part of the study, but at least the authors should discuss this limitation. Ideally they could present some comparisons with results using ERA-5 or ERA-INTERIM for some points with data only for the more recent years.

> This is a good point. We have conducted a sensitivity analysis for one site, comparing the results obtained using the 20th Century Reanalysis with ERA-5. The weather patterns driving storm surges around the UK coast are relatively large, and we found that they are captured appropriately in the coarser resolution 20th Century reanalysis. Clearly the ERA-5 captures the finer scale storm characterises better, but the coarser resolution reanalysis still reproduces the key large-scale features that are driving the compounding processes. Because the weather patterns found are relatively large scale, and captured well by the coarser reanalysis, we feel a longer time series is more beneficial than a finer temporal resolution, particularly as our analysis focuses on extreme events, which by definition are rare. In the interest of keeping the paper streamlined we won't include this in the paper and just refer to our reasons for using 20th century reanalysis. However, we are currently working on a follow up paper considering a greater number of variables, and will probably include the results of the sensitivity analysis in that second paper.

There is a lot of space for improvement regarding the figures. Before publication the authors should try to improve by removing the tables, label subplots with a, b, c...

> We were a bit confused by this comment. We have checked each figure and all subplots had already been labelled. We will increase the text sizes, to make these clearer. We included the figures in the word document in tables, for formatting reasons, but these tables will be removed from boxes for the final version.

The literature review could be updated further, for example paragraph one of the introduction omits several recent papers beyond 2017.

> Thank you for this comment. We point out that we did in fact reference at least 5 papers from 2018 (e.g. add). We are not aware of any of any papers since, but are happy to add them if the reviewer can specifically tell us which papers we are missing. In regards to paragraph 1, we included the following more recent studies:

1. Brown, S., Nicholls, R. J., Goodwin, P., Haigh, I. D., Lincke, D., Vafeidis, A. T., & Hinkel, J. (2018). Quantifying Land and People Exposed to Sea-Level Rise with no Mitigation and 1.5∘C and 2.0∘C Rise in Global Temperatures to Year 2300, *Earth's Future*, *6*. https://doi.org/10.1002/2017EF000738

2. Nerem, R.S., Beckley, B.D., Fasullo, J.T., Hamlington, B.D., Masters, D. and Mitchum, G.T., 2018. Climate-change–driven accelerated sea-level rise detected in the altimeter era. Proceedings of the National Academy of Sciences, 115(9), pp.2022-2025.

Reviewer 2 (Daniel Bachmann)

Dear authors, thanks for the very interesting paper. In the following you will find my comments, suggestions and opinion to your contribution. Summary: The paper "Assessing the characteristics and drivers of compound flooding events around the UK coast" is about a statistical analysis of compound flooding events (mainly storm surges and river flooding) for the UK. A broad data-set about high storm surge water levels and river discharges are statistically analysed. Additionally, meteorological data and data about the river catchment are included into the analysis. Three main hypotheses are tested: -Map the spatial dependency between storm surges and high river discharges -Map meteorological conditions which drive compound / non-compound events -Map dependency of catchment characteristics and compound / non-compound events All three hypotheses are well confirmed by the results of the statistical analyses. Focusing on hydrological and storm surge events the contribution addresses a relevant topic, which matches the scope of the envisaged journal HESS very well (e.g. study of the spatial and temporal characteristics of the global water resources). General remarks/suggestions to style and language -Note: I am not capable for any review about the English language because I am not a native speaker. For me, the text is comprehensible. -Sometimes it is confusing, which figure is relevant, which is for the supplement etc. Maybe check if really all tables, figures etc. are required.

> We have carefully gone through the manuscript again, and do think all of the main figures are necessary, and so there aren't any we feel can be removed. We've taken your point below about removing the data from P13L18 and P14L16 and insert into a table to reduce some of the confusion.

General remarks to content -A lot of the statements requires very good knowledge of the British geography, e.g. "...closest to the tide gauges of Bournemouth, Devonport, Workington, Ullapool, Whitby and Cromer". I excuse my ignorance of the British geography, but I cannot reproduce some of the given statements with help of the figures. I have also no perfect solution, just proposals: Give numbers to (at least) the coastal gauge stations; make detailed figures for some statements; try to focus on special stations etc.

> Thank you for this useful point. We will add tide gauge numbers to Figure 1a. Any time a tide gauge site is referred to in the text, we will add a site number in brackets.

-I missed a bit the focus to one important questions, which arise from the risk analysis / management point of view: does these compound events produce additional damages, because of their compound occurrence? It is stated in the introduction: e.g "Zscheischler et al. (2018) define compound events as the combination of multiple drivers and/or hazards that contributes to societal or environmental risk." Additionally, some examples are given from the US and the Adriatic coast. But I miss a bit more focus on the UK coast and the events which were analysed in the following chapters. In the discussion section this question is partly discussed; It comes very suddenly and unexpectedly. I suggest to make it much more prominent at the beginning of the paper (at least some remarks in the introduction). Do we observe additional damages due to compound events at the UK coast?

The reviewer raises a good point here. We have extended the introduction to make this point clearer. For example we the following sentences will be moved from the discussion to the introduction to explicitly demonstrate the compound risk around the UK.

*" The 24th-25th December 1999 compound flooding event in Lymington is especially noteworthy and illustrates the need to consider compound events in the design of flood protection schemes. On the 16-17th December 1989 Lymington was flooded by high sea levels and waves, with considerable damage to 50 houses and the railway line (Ruocco et al., 2011; Haigh et al., 2015). This event was the driving force for a large upgrade of coastal flood defences for the town, including new sluice gates which allowed the Lymington River to drain at low tide, but sealed it from tidal flooding during high sea levels. However, no allowance or consideration of compound flooding appears to have been made at the time. Ten years later, on 24th December 1999, a storm generated a storm surge which did not directly cause flooding itself, because of the raised defences. However, the storm surge prevented the sluice gates from opening for prolonged periods, while large volumes of rainfall during the storm raised river flow. Combined with the lack of drainage, this caused flooding from the river on the upstream side of the sea defences (Ruocco et al., 2011). Subsequently the Lymington flood defences were upgraded again. This event strongly highlights the importance of considering compound flooding when assessing and designing flood management. "*

A second question is, why "just extreme events" were analysed. It is also stated in the introduction: "combinations of events that are not themselves extremes but lead to an extreme event when combined"! I think especially these events are the important events which we've miss until now in risk analysis. Maybe make a remark that this point is excluded from the analysis (if it is not done somewhere).

Within this paper we have focused on just the extreme events. We agree that combinations of moderate events are important, however, it's beyond the scope of this paper to include them in the analysis. We've added a short paragraph in the discussion to highlight the importance of these events and need to analyse them.

*" As stated earlier, compound flooding can occur not only during 2 (or more) extreme events, but also when just one flood source is extreme (for example, extreme river discharge combines with a moderate storm surge); or when two moderate flooding sources combine to create a flood event. It should be noted, that the latter two types of compound flooding involving moderate events were beyond the scope of this paper and so were not considered in our methods. These types of events are important however and need to be recognised in future studies into flood risk. "*

-I would like to note a very similar remark about the discussion of the forecasting strategies also in the "discussion"-chapter. It comes also very suddenly and unexpectedly in this chapter. I think it could be also (at least) mentioned in the introduction.

We agree with this point of not introducing the forecasting earlier. We shall mention the forecasting element in the introduction, by moving the following paragraph from P19 L18 to the introduction

*"The UK Flood Forecasting Centre (a collaboration between the Environment Agency and Met Office) have developed a medium- to long-range operational forecasting tool called Coastal Decider (Neal et al., 2018). This is based on probabilistic weather-pattern forecasts and helps to identify periods with an increased likelihood of coastal flooding from high sea levels around the UK. Coastal Decider uses a set of 30 distinct weather patterns (referred to*

*as the 'Met Office weather patterns') which were derived by Neal et al. (2016) using k-means clustering techniques. These weather patterns (shown in Supplementary Figure S32) represent the large-scale meteorological conditions experienced over the UK and surrounding European area. Neal et al. (2018) used a daily historical weather pattern catalogue to show that particular weather patterns tend to relate to high sea level events at different sites around the UK, with this analysis forming the basis for Coastal Decider. Other research which relates the Met Office weather patterns to meteorologically induced hazards includes Richardson et al. (2018), who related the weather patterns to precipitation observations for the application of drought forecasting."*

-Sometimes I would prefer that the hypotheses are more feed with physical reasoning, especially when the results are described: in some cases, it is done, in other cases it is done in the "discussion"-chapter, sometimes it could be a bit more pronounced. The question is: just give the physical explanation in the "discussion"-chapter or immediately after the description of the results? In this case, where a lot of results are described a direct explanation could be preferable (suggestions!). If the explanations first come in the discussion-chapter, some important details are already forgotten by the reader. Moreover, the purpose of a discussion chapter is more (as far as I know): discuss your findings critically, what are the limitations?, what could be better?, in which direction will it go? Etc.

> We understand the reviewers point, however we have deliberately written up this study in traditional format, with the results and the interpretation of results explicitly separated. We would prefer to keep the paper in this format, particularly as this study has been carried out as part of a PhD and will later be incorporated into a final thesis.

Finally, I have a bit the feeling, that too much information (esp. results) is given to the reader. Therefore, much more detailed descriptions are sometimes lacking. Maybe a review of important information (text / figures etc.) should be done. Detailed Comments and proposals. See attached document. Conclusion of the review. The presented article is a very interesting contribution about the statistical analysis of compound flood events. I am convinced that this paper is a valuable contribution to the journal HESS. I hope my suggestions, remarks, comments can help to improve this manuscript further. Please also note the supplement to this comment: https://www.hydrol-earth-syst-sci-discuss.net/hess-2018-632/hess-2018-632-RC2- supplement.pdf

P4 L8: Suggestion:

Make the UK coast and their compound events and their additional damages more prominent (see general remarks to content)

> The Lymington event described in the results will be moved to the introduction to highlight compound flooding is in an issue in the UK, however the examples documented causing the most damage so far have been non-UK based and so are useful to highlight why compound flooding is such an important issue.

P6 L14 :

This is quite short; is it really required?

> We agree with the reviewer, the layout of the paper is not required, it will be removed.

P6L24: Do you use measured and astronomical predicted values?

> We use measured sea levels and predicted astronomical tides (using T-Tide package) to calculate the skew surge. This has been described in the methods (Section 3.1).

P7L5 : How they are selected: One gauge station for one "whole" river catchments (from source to mouth)?

Are sub catchments also used?

The site selection is described in Section 2.5 based on:

(1) there are at least 15 years of overlapping records; and
(2) daily mean river discharge is at least 5 $m^3$/s at the river site.

These have been matched to the nearest tide gauge that is hydrologically-relevant, i.e. the mouth is near the river mouth. Sub-catchment have been used, sometimes with multiple river gauges on the same river have been used. This point will be clarified in Section 2.5.

P7L28 : Could you give a formular?

In the interest of keeping the paper concise we will include the formula in the supplementary material rather than the main paper.

P7L32: This is a direct reference to the NRFA web-page! Could you mark it?

Thank you for pointing this out, we shall include the reference.

P8L15: Did they do any final visual check, if the combination make any sense? Because a first automatic combination, with an final visual check, should be the best method, isn't it?

The studies referred to (Paprotny et al 2018 and Svensson and Jones, 2002 and 2004), did not visually check if the combinations of river and tide gauges made sense. Most likely this is due to the large spatial scale involved in their studies. In our study, we have made sure the river mouth of each river station is near to the tide gauge. We'll specify which the studies we're referring to in this sentence.

P8L19: Are the names really important here? (see also general remarks to content)

Whilst we understand the reviewers point that this isn't useful for readers unfamiliar with UK geography, we feel it could be useful to UK based readers in understanding specific locations.

P10L5: Is "extreme" enough for compound events? (see general remarks to the content)

As mentioned earlier, we agree that combinations of moderate events are important, however, it's beyond the scope of this paper to include them in the analysis. We shall mention this as a limitation of the study in the discussion section.

P10L14: p (italic)?

We will correct this and change it to italics.

P10L22: Is a word missing here?

We will reword to *"The second approach we term hereafter the 'dependence extreme method' and it follows the methods previously applied by, for example, Wahl et al. (2015) and Ward et al. (2018)."*

P10L26: Maybe mark it as one expression with italic letters or "...."(see also below)

> We will refer to it in italics as *"discharge conditional on high total sea-level or skew surge"* throughout the paper to be clearer.

P11L7 : Why not using the total maximum surge? From the damage / risk point of view this one is relevant!

> During our analysis we used both skew surge and total water level. As shown, when comparing Figures 3 and 4, the relationship is far stronger when using the skew surge due to removing the astronomical tidal influence. We have therefore only included the skew surge in the subsequent meteorological analysis to best show physical mechanisms linking the surge and river discharge.

P13L15: What is the reason for this observation? I know it is stated in the discussion, but this is still some pages away (see also general remarks to content)

> As discussed above we've deliberately written the paper in this style with results purely a description of the analysis and the discussion to explain them.

P13L18: Could this be summarized in a table?1

> The percentages of sites experiencing maximum correlation and joint occurrences will be summarized in a table to reduce the wording and confusion in the text.

P13L26 : To much information??

> We agree with the reviewer, the comparison of two methods of dependence calculations are not necessary and clutter the results, we shall remove this paragraph from the results and the figures from the supplementary material.

P14L9: UK geography (see also general remarks to content)

> These sites will also be numbered for those less familiar with UK geography.

P14L16: Put this in a Table?

> As with comment on P13L18 this will be put in table and referred to.

P15L3: These composite plots are derived for (almost) whole Europe, isn't it?

> The derived area is specified in Section 2.3, as this is the region where storms affecting the UK are typically generated/influence the region.

P15L4: Repetition from the "method"-chapter

> We will simplify this to *"At each of the 326 river discharge sites, we have derived composite plots of SLP, wind speed and PWC through the time of the events that have led to:"* so there is some reminder to the reader what we have done, but not too much repetition.

P16L19: Maybe formulated it a bit more moderate: also the upper panels shows some differences. The direction is in all cases the same, isn't it?

We'll include some reasoning for the differences, most likely due to the resolution of the average storm track.

Direction is the same for all tracks (west to east), we shall clarify this.

P16L25: ("above" not required)

"above" shall be removed

P16L33: Is it true for this area? Than write it!

We have altered this to *"The river sites that drain onto the central south coast typically have the greatest BFI (nearly 1, i.e. extremely porous chalk), whilst those on the north west coast typically have the lowest (0-0.2, i.e. predominately clay soils) (Figure 10a)."*

P17L7: this is really difficult to judge just based on the picture. For example: Shouldn't be any correlation between steepness and the catchments area? Do not smaller catchments tend to be steeper? Or has the huge river Rhine catchments also a large steepnes factor, because it goes from 0 mNN (mouth) up to 2300 mNN (source)?

The elevation variation is calculated from the $10^{th}$ and $90^{th}$ percentiles of each catchment's elevation, as opposed the minimum and maximum. This therefore captures the profile in terms of steepness, rather than which than just which has the maximum elevation.

P17L26: "catchment"? or characteristic?

We have changed this to *"catchment characteristics"*.

P18L31: Starting from here very important points are stated. Maybe parts of it should be at the beginning of the paper. (see also general remarks to content)

We will ensure the important points of the paper (such as the lag between surge and river being a key concern to flooding) are mentioned in the introduction.

P22L11: Therefore it is important to investigate, if additional damages occur in case of a compound event!

We shall expand the final sentence to *"The previous lack of consideration of compound flooding means that flood risk has likely been underestimated around UK coasts, particularly along the southwest and west coasts. Furthermore, the additional damages caused due to compound events are unknown. It is crucial that this is addressed in future assessments of flood risk and flood management approaches"*.

P29 Fig3: This is really difficult to recognize on the figure? (bold circles)

This a very good point, as most sites are statistically significant, it's hard to spot those which aren't. We will instead mark the sites that aren't significant with hatching.